# EpidemiOptim: a toolbox for the optimization of control policies in epidemiological models

## Abstract

Epidemiologists model the dynamics of epidemics in order to propose control strategies based on pharmaceutical and non-pharmaceutical interventions (contact limitation, lock down, vaccination, etc). Hand-designing such strategies is not trivial because of the number of possible interventions and the difficulty to predict long-term effects. This task can be cast as an optimization problem where state-of-the-art machine learning algorithms such as deep reinforcement learning, might bring significant value. However, the specificity of each domain – epidemic modelling or solving optimization problem – requires strong collaborations between researchers from different fields of expertise. This is why we introduce EpidemiOptim, a Python toolbox that facilitates collaborations between researchers in epidemiology and optimization. EpidemiOptim turns epidemiological models and cost functions into optimization problems via a standard interface commonly used by optimization practitioners (*OpenAI Gym*). Reinforcement learning algorithms based on Q-Learning with deep neural networks (DQN) and evolutionary algorithms (NSGA-II) are already implemented. We illustrate the use of EpidemiOptim to find optimal policies for dynamical on-off lock-down control under the optimization of death toll and economic recess using a Susceptible-Exposed-Infectious-Removed (SEIR) model for COVID-19. Using EpidemiOptim and its interactive visualization platform in Jupyter notebooks, epidemiologists, optimization practitioners and others (e.g. economists) can easily compare epidemiological models, costs functions and optimization algorithms to address important choices to be made by health decision-makers.

## 1 Introduction

The recent COVID-19 pandemic highlights the destructive potential of infectious diseases in our societies, especially on our health, but also on our economy. To mitigate their impact, scientific understanding of their spreading dynamics coupled with methods quantifying the impact of intervention strategies along with their associated uncertainty, are key to support and optimize informed policy making. For example, in the COVID-19 context, large scale population lock-downs were enforced based on analyses and predictions from mathematical epidemiological models (Ferguson et al., 2005; 2006; Cauchemez et al., 2019; Ferguson et al., 2020). In practice, researchers often consider a small number of relatively coarse and pre-defined intervention strategies, and run calibrated epidemiological models to predict their impact (Ferguson et al., 2020). This is a difficult problem for several reasons: 1) the space of potential strategies can be large, heterogeneous and multi-scale (Halloran et al., 2008); 2) their impact on the epidemic is often difficult to predict; 3) the problem is multi-objective by essence: it often involves public health objectives like the minimization of the death toll or the saturation of intensive care units, but also societal and economic sustainability. For these reasons, pre-defined strategies are bound to be suboptimal. Thus, a major challenge consists in leveraging more sophisticated and adaptive approaches to identify optimal strategies.

Machine learning can be used for the optimization of such control policies, with methods ranging from deep reinforcement learning to multi-objective evolutionary algorithms. In other domains, they have proven efficient at finding robust control policies, especially in high-dimensional non-stationary environments with uncertainty and partial observation of the state of the system (Deb et al., 2007; Mnih et al., 2015; Silver et al., 2017; Haarnoja et al., 2018; Kalashnikov et al., 2018; Hafner et al., 2019). Yet, researchers in epidemiology, in public-health, in economics, and in ma-

chine learning evolve in communities that rarely cross, and often use different tools, formalizations and terminologies. We believe that tackling the major societal challenge of epidemic mitigation requires interdisciplinary collaborations organized around operational scientific tools and goals, that can be used and contributed by researchers of these various disciplines. To this end, we introduce EpidemiOptim, a Python toolbox that provides a framework to facilitate collaborations between researchers in epidemiology, economics and machine learning.

EpidemiOptim turns epidemiological models and cost functions into optimization problems via the standard OpenAI Gym (Brockman et al., 2016) interface that is commonly used by optimization practitioners. Conversely, it provides epidemiologists and economists with an easy-to-use access to a variety of deep reinforcement learning and evolutionary algorithms, capable of handling different forms of multi-objective optimization under constraints. Thus, EpidemiOptim facilitates the independent update of models by specialists of each topic, while enabling others to leverage implemented models to conduct experimental evaluations. We illustrate the use of EpidemiOptim to find optimal policies for dynamical on-off lock-down control under the optimization of death toll and economic recess using an extended Susceptible-Exposed-Infectious-Removed (SEIR) model for COVID-19 from Prague et al. (2020).

**Related Work.** We can distinguish two main lines of contributions concerning the optimization of intervention strategies for epidemic response. On the one hand, several contributions focus on providing guidelines and identifying the range of methods available to solve the problem. For example, Yáñez et al. (2019) framed the problem of finding optimal intervention strategies for a disease spread as a reinforcement learning problem; Alamo et al. (2020) provided a road-map that goes from the access to data sources to the final decision-making step; and Shearer et al. (2020) highlighted that a decision model for epidemic response cannot capture all of the social, political, and ethical considerations that these decisions impact. These contributions reveal a major challenge for the community: developing tools that can be easily used, configured and interpreted by decision-makers. On the other hand, computational contributions proposed actual concrete implementations of such optimization processes. These contributions mostly differ by their definition of epidemiological models (e.g. SEIR (Yaesoubi et al., 2020) or agent-based models (Chandak et al., 2020)), of optimization methods (e.g. deterministic rules (Tarrataca et al., 2020), Bayesian optimization (Chandak et al., 2020), Deep RL (Arango & Pelov, 2020) or evolutionary optimization (Miikkulainen et al., 2020)), of cost functions (e.g. fixed weighted sum of health and economical costs (Arango & Pelov, 2020), possibly adding constraints on the school closure budget (Libin et al., 2020), or multi-objective optimization (Miikkulainen et al., 2020)), of state and action spaces (e.g. using the entire observed epidemic history (Yaesoubi et al., 2020) or an image of the disease outbreak to capture the spatial relationships between locations (Probert et al., 2019)), as well as methods for representing the model decisions in a format suitable to decision-makers (e.g simple summary state representation (Probert et al., 2019) or real-time surveillance data with decision rules (Yaesoubi & Cohen, 2016)). See Appendix A.2 for a detailed description of the aforementioned papers.

Given this high diversity of potential methods in the field, our approach aims at providing a standard toolbox facilitating the comparison of different configurations along the aforementioned dimensions in order to assist decision-makers in the evaluation of the range of possible intervention strategies.

**Contributions.** This paper makes three contributions. First, we formalize the coupling of epidemiological models and optimization algorithms with a particular focus on the multi-objective aspect of such problems (Section 2). Second, based on this formalization, we introduce the EpidemiOptim library, a toolbox that integrates epidemiological models, cost functions, optimization algorithms and experimental tools to easily develop, study and compare epidemic control strategies (Section 3). Third, we demonstrate the utility of the EpidemiOptim library by conducting a case study on the optimization of lock-down policies for the COVID-19 epidemic (Section 4). We use a recent epidemiological model grounded on real data, cost functions based on a standard economical model of GDP loss, as well as state-of-the-art optimization algorithms, all included in the EpidemiOptim library. This is, to our knowledge, the first contribution that provides a comparison of different optimization algorithm performances for the control of intervention strategies on the same epidemiological model. The user of the EpidemiOptim library can interact with the trained policies via Jupyter notebook, exploring the space of cost functions (health cost x economic cost) for a variety of algorithms. The code is made available anonymously at `https://tinyurl.com/epidemioptim`.

## 2  THE EPIDEMIC CONTROL PROBLEM AS AN OPTIMIZATION PROBLEM

In an *epidemic control problem*, the objective is to find an optimal control strategy to minimize some cost (e.g. health and/or economical cost) related to the evolution of an epidemic. We take the approach of reinforcement learning (RL) (Sutton & Barto, 2018): the control policy is seen as a *learning agent*, that interacts with an epidemiological model that is seen as its *learning environment*, see Figure 1. Each run of the epidemic is a *learning episode*, where the agent alternatively interacts with the epidemic through *actions* and observes the resulting *states* of the environment and their associated *costs*. To define this optimization problem, we define three elements: 1) the state and action spaces; 2) the epidemiological model; 3) the cost function.

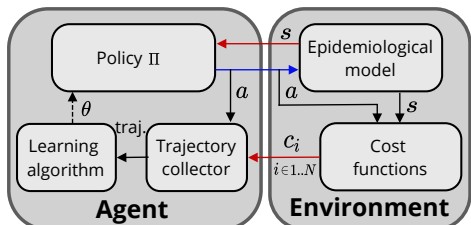

Figure 1: **Epidemic control as an optimization problem.** $s, a, c_i$ refer to environment states, control actions, and the $i^{th}$ cost, while *traj.* is their collection over an episode. Blue and red arrows match the input and output of the OpenAI Gym *step* function.

**State and action spaces.**   The state $s_t$ describes the information the control policy can use (e.g. epidemiological model states, cumulative costs up to timestep $t$, etc.) We also need to define which actions $a$ can be exerted on the epidemic, and how they affect its parameters. For instance, enforcing a lock-down can reduce the transmission rate of the epidemic and slow it down.

**The epidemiological model.**   The epidemiological model is akin to a *transition function* $\mathcal{T}$: it governs the evolution of the epidemic. From the current state $s_t$ of the epidemic and the current action $a_t$ exerted by the agent, it generates the next state $s_{t+1}$: $\mathcal{T} : \mathcal{S}, \mathcal{A} \to \mathcal{S}$ where $\mathcal{S}, \mathcal{A}$ are the state and action spaces respectively. Note that this mapping can be stochastic.

**Cost functions and constraints.**   The design of cost functions is central to the creation of optimization problems. Epidemic control problems can often be framed as multi-objective problems, where algorithms need to optimize for a set of cost functions instead of a unique one. Costs can be defined at the level of timesteps $C(s, a)$ or at the level of an episode $C_e(\text{trajectory})$. This cumulative cost measure can be obtained by summing timestep-based costs over an episode. For mathematical convenience, RL practitioners often use a discounted sum of costs, where future costs are discounted exponentially: $C_e(\text{trajectory}) = \sum_t \gamma^t c(s_t, a_t)$ with discount factor $\gamma$ (e.g. it avoids infinite sums when the horizon is infinite). When faced with multiple cost functions $C_i|_{i \in [1..N_c]}$, a simple approach is to aggregate them into a unique cost function $C$, computed as the convex combination of the individual costs: $\bar{C}(s, a) = \sum_i \beta_i C_i(s, a)$. Another approach is to optimize for the Pareto front, i.e. the set of non-dominated solutions. A solution is said to be non-dominated if no other solution performs better on all costs and strictly better on at least one. In addition, one might want to define constraints on the problem (e.g. a maximum death toll over a year).

**The epidemic control problem as a Markov Decision Process.**   Just like traditional RL problems, our epidemic control problem can be framed as a Markov Decision Process: $\mathcal{M} : \{\mathcal{S}, \mathcal{A}, \mathcal{T}, \rho_0, C, \gamma\}$, where $\rho_0$ is the distribution of initial states, see Yáñez et al. (2019) for a similar formulation.

**Handling time limits.**   Epidemics are not episodic phenomena, or at least it is difficult to predict their end in advance. Optimization algorithms, however, require episodes of finite lengths. This enables the algorithm to regularly obtain fresh data by trying new policies in the environment. This can be a problem, as we do not want the control policy to act as if the end of the episode was the end of the epidemic: if the epidemic is growing rapidly right before the end, we want the control policy to care and react. Evolution algorithms (EAs) cannot handle such behavior as they use episodic data $(\theta_e, C_e)$, where $\theta_e$ are the policy parameters used for episode $e$. RL algorithms, however, can be made time unaware. To this end, the agent should not know the current timestep, and should not be made aware of the last timestep (termination signal). Default RL implementations often ignore this fact, see a discussion of time limits in Pardo et al. (2018).

# 3 THE EPIDEMIOPTIM TOOLBOX

## 3.1 TOOLBOX DESIDERATA

The EpidemiOptim toolbox aims at facilitating collaborations between the different fields interested in the control of epidemics. Such a library should contain state-of-the-art epidemiological models, optimization algorithms and relevant cost functions. It should enable optimization practitioners to bring their newly-designed optimization algorithm, to plug it to interesting environments based on advanced models grounded in epidemiological knowledge and data, and to reliably compare it to other state-of-the-art optimization algorithms. On the other hand, it should allow epidemiologists to bring their new epidemiological model, reuse existing cost functions to form a complete epidemic control problem and use state-of-the-art optimization algorithms to solve it. This requires a modular approach where algorithms, epidemiological models and cost functions are implemented independently with a standardized interface. The toolbox should contain additional tools to facilitate experiment management (result tracking, configuration saving, logging). Good visualization tools would also help the interpretation of results, especially in the context of multi-objective problems. Ideally, users should be able to interact with the trained policies, and to observe the new optimal strategy after the modification of the relative importance of costs in real-time. Last, the toolbox should enable reproducibility (seed management) and facilitate statistical comparisons of results.

## 3.2 TOOLBOX ORGANIZATION

**Overview.**   The EpidemiOptim toolbox is centered around two modules: the *environment* and the *optimization algorithm*, see Figure 1. To run an experiment, the user can define its own or select the ones contained in the library. Appendix Figure 4 presents the main interface of the EpidemiOptim library. The following sections delve into the *environment* and *algorithm* modules.

**Environment: epidemiological models.**   The *Model* module wraps around epidemiological models, and allows to sample new model parameters from a distribution of models, to sample initial conditions ($\rho_0$) and to run the model for $n$ steps. Currently, the EpidemiOptim library contains a Python implementation of an extended Susceptible-Exposed-Infectious-Removed (SEIR) model fitted to French data, from Prague et al. (2020).

**Environment: cost functions and constraints.**   In the EpidemiOptim toolbox, each cost function is a separate class that can compute $c_i(s, a)$, normalize the cost to $[0, 1]$ and generate constraints on its cumulative value. The *Multi-Cost* module integrates several cost functions. It computes the list of costs $c_i(s, a)|_{i \in [1..N_c]}$ and an aggregated cost $\bar{c}$ parameterized by the mixing weights $\beta_i|_{i \in [1..N_c]}$. The current version of EpidemiOptim contains two costs: 1) a health cost computed as the death toll of the epidemic; 2) an economic cost computed as the opportunity loss of GDP resulting from either the epidemic itself (illness and death reduce the workforce), or the control strategies (lockdown also reduces the workforce). In addition, we define two constraints as maximal values that the cumulative measures of the two costs described above can take.

**Environment: OpenAI Gym interface as a universal interface.**   The learning environment defines the optimization problem. We use a framework that has become standard in the Reinforcement Learning community in recent years: the *OpenAI Gym environment* (Brockman et al., 2016). Gym environments have the following interface: First, they have a *reset* function that resets the environment state, effectively starting a new simulation or episode. Second, they have a *step* function that takes as input the next action from the control policy, updates the internal state of the model and returns the new environment state, the cost associated to the transition and a Boolean signal indicating whether the episode is terminated, see Figure 1. This standardized interface allows optimization practitioners (RL researchers especially) to easily plug any algorithm, facilitating the comparison of algorithms on diverse benchmark environments. In our library, this class wraps around an epidemiological model and a set of cost functions. It also defines the state and action spaces $\mathcal{S}$, $\mathcal{A}$. So far, it contains the *EpidemicDiscrete-v0* environment, a Gym-like environment based on the epidemiological model and the bi-objective cost function described above. Agents decide every week whether to enforce a partial lock-down (binary action) that results in a decreased transmission rate.

**Optimization algorithms.** The optimization algorithm, or *algorithm* for short, trains learning agents to minimize cumulative functions of costs. Learning agents are characterized by their control policy: a function that takes the current state of the environment as input and produces the next action to perform: $\Pi_\theta : \mathcal{S} \rightarrow \mathcal{A}$ where $\Pi_\theta$ is parameterized by the vector $\theta$. The objective of the algorithm is, thus, to find the optimal set of parameters $\theta^*$ that minimizes the cost. These algorithms collect data by letting the learning agent interact with the environment. Currently, the EpidemiOptim toolbox integrates several RL algorithms based on the famous Deep Q-Network algorithm (Mnih et al., 2015) as well as the state-of-the-art multi-objective EA NSGA-II Deb et al. (2002).

**Utils.** We provide additional tools for experimentation: experience management, logging and plots. Algorithms trained on the same environment can be easily compared, e.g. by plotting their Pareto front on the same graph. We also enable users to visualize and interact with trained policies via Jupyter notebooks. Users can select solutions from the Pareto front, modify the balance between different cost functions and modify the value of constraints set to the agent via sliders. They visualize a run of the resulting policy in real-time. We are also committed to reproducibility and robustness of the results. For this reason, we integrate to our framework a library for statistical comparisons designed for RL experiments (Colas et al., 2019).

## 4 CASE STUDY: LOCK-DOWN ON-OFF POLICY OPTIMIZATION FOR THE COVID-19 EPIDEMIC

The SARS-CoV-2 virus was identified on January, 7th 2020 as the cause of a new viral disease named COVID-19. As it rapidly spread globally, the World Health Organization (2020) declared a pandemic on March 11th 2020. Exceptional measures have been implemented across the globe as an attempt to mitigate the pandemic (Kraemer et al., 2020). In many countries, the government response was a total lock-down: self-isolation with social distancing, schools and workplaces closures, cancellation of public events, large and small gatherings prohibition, and travel limitation. In France, lock-down lasted 55 days from March, 17th to May, 11th. It led to a decrease in gross domestic product of, at least, 10.1% (Mandel & Veetil, 2020) but allowed to reduce the death toll by an estimated 690,000 [570,000; 820,000] people (Flaxman et al., 2020). In this case study, we investigate alternative strategies optimized by RL and EA algorithms over a period of 1 year.

### 4.1 ENVIRONMENT

**The epidemiological model.** We use a mathematical structural model to understand the large-scale dynamics of the COVID-19 epidemic. We focus on the extended Susceptible-Exposed-Infectious-Removed (SEIR) model from Prague et al. (2020), which was fitted to French data. The estimation in this model was region-specific. Here, we use their estimated parameters for the epidemic dynamics in *Île-de-France* (Paris and its surrounding region), presented in Appendix Table 1. The model explains the dynamics between Susceptibles ($S$), Exposed ($E$), Ascertained Infectious ($I$), Removed ($R$), Non-Ascertained Asymptomatic ($A$) and Hospitalized ($H$). The underlying system of differential equations and a schematic view of the mechanistic model can be found in Appendix Section A.3. To account for uncertainty in the model parameters, we create a distribution of models resulting from parameter distributions, see details in Appendix Section A.3 and a visualization of 10 sampled models in Appendix Section A.3.2.

**State and action spaces.** The state of the agent is composed of the epidemiological states (SEIRAH) normalized by the population size ($N = 12,278,210$), booleans indicating whether the previous and current state are under lock-down, the two cumulative costs and the level of the transmission rate (from 1 to 4). The two cumulative costs are normalized by $N$ and $150B$ respectively, $150B$ being the approximate maximal cost due to a full year of lockdown. The action occurs weekly and is binary (1 for lock-down, 0 otherwise). Consecutive weeks of lockdown decrease the $b$ in four stages (see model description in Appendix Figure 5), while stopping lockdown follows the opposite path (stair-case increase of $b$).

**Cost functions.** In this case-study, an optimal policy should minimize two costs: a health cost and an economic cost. The health cost is computed as the death toll, it can be evaluated for each environment transition (weekly) or over a full-episode (e.g. annually): $C_h(t) = D(t) = 0.005R(t)$.

The economic cost is measured in euros as the GDP opportunity cost resulting from diseased or dead workers, as well as people unemployed due to the lock-down policy. At any time $t$, the GDP can be expressed as a Cobb-Douglas function $F(L(t)) = AK_0^{\gamma_k} L(t)^{1-\gamma_k}$, where $K_0$ is the capital stock of economy before the outbreak that we will suppose constant during the pandemics, $L(t)$ is the active population employed, $\gamma_k$ is the capital elasticity and $A$ is the exogenous technical progress (Douglas, 1976). Let $L_0 = \lambda N$ be the initial employed population, where $\lambda$ is the activity/employment rate. The lock-down leads to partial unemployment and decreases the size of the population able to work (illness, isolation, death), thus $L(t) = (1 - u(t))\lambda(N - G(t))$, where $u(t)$ is the level of partial unemployment due to the lock-down and $G(t) = I(t) + H(t) + 0.005R(t)$ is the size of the ill, isolated or dead population as defined in our SEIRAH model. The economic cost is defined as the difference between the GDP before ($Y_0$) and after the pandemic, $Y_0 - F(L(t))$ and is given by:

$$C_{\text{eco}}(t) = Y_0 - AK_0^{\gamma_k}((1 - u(t))\lambda(N - G(t))^{1-\gamma_k}$$

Parameters for values of parameters in the economic model are given in Table 1. We also consider the use of constraints on the maximum value of cumulative costs.

## 4.2 Optimization Algorithms

We consider three algorithms: 1) a vanilla implementation of Deep Q-Network (DQN) (Mnih et al., 2015); 2) a goal-conditioned version of DQN inspired from (Schaul et al., 2015); 3) NSGA-II, a state-of-the-art multi-objective evolutionary algorithm (Deb et al., 2002). The trained policy is the same for all algorithms: a neural network with one hidden-layer of size $64$ and ReLU activations. Further background about these algorithm is provided in Appendix SectionA.3.

**Optimizing convex combinations of costs: DQN and Goal-DQN.** DQN only targets one cost function: $\bar{c} = (1 - \beta) c_{\text{h}} + \beta c_{\text{eco}}$. Here the health and economic costs are normalized to have similar ranges (scaled by $1/(65 \times 1e3)$ and $1/1e9$ respectively). We train independent policies for each values of $\beta$ in $[0., 0.05, .., 1]$. This consists in training a Q-network to estimate the value of performing any given action $a$ in any given state $s$, where the value is defined as the cumulative negative cost $-\bar{c}$ expected in the future. Goal-DQN, however, trains one policy $\Pi(s, \beta)$ to target any convex combination parameterized by $\beta$. We use the method presented in (Badia et al., 2020): we train two Q-networks, one for each of the two costs. The optimal action is then selected as the one maximizing the convex combination of the two Q-values:

$$\Pi(s, \beta) = \text{argmax}_a (1 - \beta) Q_{\text{s}}(s, a) + \beta Q_{\text{eco}}(s, a). \tag{1}$$

By disentangling the two costs, this method facilitates the representation of values that can have different scales and enables automatic transfer for any value of $\beta$, see justifications in Badia et al. (2020). During training, agents sample the targeted $\beta$ uniformly in $[0, 1]$.

**Adding constraints with Goal DQN-C.** We design and train a variant of Goal DQN to handle constraints on maximal values for the cumulative costs: Goal-DQN-C. We train a Q-network for each constraints with a cost of 1 each time the constraint is violated, 0 otherwise ($Q_{\text{h}}^c, Q_{\text{eco}}^c$). With a discount factor $\gamma = 1$, this network estimates the number of transitions that are expected to violate the constraints in the future. The action is selected according to Eq. 1 among the actions that are not expected to lead to constraints violations ($Q^c(s, a) < 1$). If all actions are expected to lead to violations, the agent selects the action that minimizes that violation. During training, agents sample $\beta$ uniformly, and $50\%$ of the time samples uniformly one of the two constraints in $[1000, 62000]$ for the maximum death toll and $[20, 160]$ for the maximum economic cost, in billions.

**EAs to optimize a Pareto front of solutions.** We also use NSGA-II, a state-of-the-art multi-objective EA algorithm that trains a population of solutions to optimize a Pareto front of non-dominated solutions (Deb et al., 2002). As others EAs, NSGA-II uses episodic data $(\theta_e, C_e)$. To obtain reliable measures of $C_e$ in our stochastic environment, we average costs over 30 evaluations. For this reason, NSGA-II requires more samples than traditional gradient-based algorithms. For fair comparison, we train two NSGA-II algorithm: one with the same budget of samples than DQN variants (1e6 environment steps), and the other with 15x as much (after convergence).

## 4.3 RESULTS

**Pareto fronts.** We aim at providing decision tools to decision-maker and, for this reason, cannot decide ourselves the right value for the mixing parameter $\beta$. Instead, we want to present Pareto fronts of Pareto-optimal solutions. Only NSGA-II aims at producing an optimal Pareto front as its result while DQN and Goal DQN optimize a unique policy. However, one can build Pareto fronts even with DQN variants. We can build population of policies by aggregating independenlty trained DQN policies, or evaluating a single Goal DQN policy on a set of $N_{\text{pareto}}$ goals sampled uniformly (100 here). The resulting population can then be filtered to produce the Pareto front. Figure 2a presents the resulting Pareto fronts for all variants. The Pareto fronts made of several DQN policies (yellow) perform better in the regime of low health cost ($< 10^3$ deaths) where it leads to save around 20 B euros over NSGA-II and the Goal DQN without constraint. Over $25 \times 10^3$ deaths, NSGA-II becomes the best algorithm, as it allows to save 10 to 15 B euros in average, the health cost being kept constant.

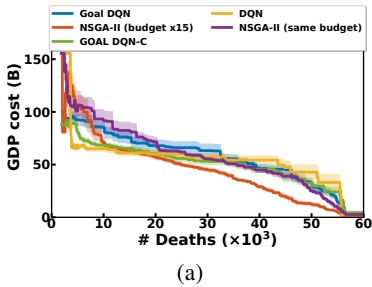
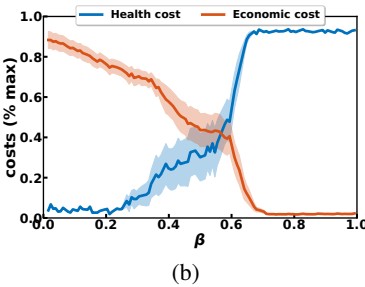

(a)  (b)

Figure 2: Left: Pareto fronts for various optimization algorithms. The closer to the origin point the better. Right: Evolution of the costs when Goal DQN agents are evaluated on various $\beta$. We plot the mean and standard error of the mean (left) and standard deviations (right) over 10 seeds.

In the low health cost regime, the DQN policy attempts to break the first wave of the epidemic, then maintains it low with periodic lock-down (1 or 2 weeks period) until it is sufficiently low. If a second wave appears, it goes back to that strategy, see Figure 3a for an example. Goal DQN with constraints shows a similar behavior. NSGA-II enters a cyclic 1-week period of lock-down but does not stop it even when the number of new cases appears null (Appendix Section A.4). In the low economic cost regime (below 15 B), NSGA-II waits for the first wave to grow and breaks it by a lock-down of a few weeks (Figure 3b). DQN-based algorithms sometimes find such policies. Policies in the range of health cost between $10^3$ and $50 \times 10^3$ deaths are usually policy that perform low or high health costs depending on the particular epidemic and its initial state. Appendix Section 4.3 delves into the different strategies and interesting aspects of the four algorithms.

**Optimizing convex combinations.** When $\beta$ is low (respectively high), the cost function is closer to the health (respectively economic) cost (see Eq 1). Figure 2b presents the evolution of the two costs when a trained Goal DQN policy (without constraints) is evaluated on different $\beta$. As expected, the higher $\beta$ the lower the health cost and vice-versa. We see that, outside of the $[0.2, 0.8]$ range, the policy falls in extreme regimes: either it always locks-down (low $\beta$) or it never does (high $\beta$).

**Interactive decision making.** The graphs presented above help make sense of the general performance of the algorithms, in average over multiple runs. However, it is also relevant to look at particular examples to understand a phenomenon. For this purpose, we include visualization Jupyter Notebooks in the EpidemiOptim toolbox. There are three notebooks for DQN, Goal DQNs and NSGA-II respectively. The two first one help making sense of DQN and Goal DQN algorithms (with or without constraints). After loading a pre-trained model, the user can interactively act on the cost function parameters (mixing weight $\beta$ or constraints on the maximum values of the cumulative costs). As the cost function changes, the policy automatically adapts and the evolution of the epidemic and control strategy are displayed. The third notebook helps visualizing a Pareto front of the NSGA-II algorithm. After loading a pre-trained model, the user can click on solutions shown on the Pareto front, which automatically runs the corresponding model and displays the evolution of the epidemic and control strategy. The user can explore these spaces and visualize the strategies that correspond to the different regimes of the algorithm.

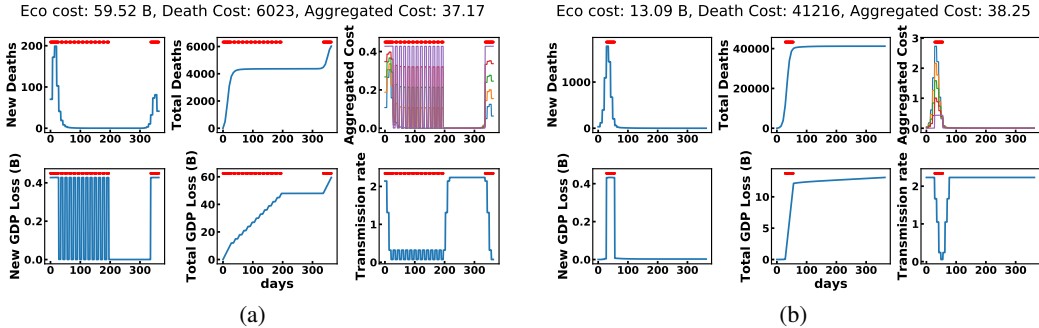

Figure 3: Evolution of cost metrics and strategy over a year of simulated epidemic with a DQN agent (left) in the low health cost regime and a NSGA-II agent in the low economic cost regime (right).

**Discussion.**    This case study proposed to use a set of state-of-the-art algorithms to target the problem of lock-down policy optimization in the context of the COVID-19 epidemic (RL and EA algorithms). DQN appears to work better in the regime of low health cost, but requires to train multiple policies independently. As it trains only one policy, Goal DQN is faster, but does not bring any performance advantage. NSGA-II, which is designed for the purpose of multi-objective optimization, seems to perform better in the regime of low economic cost.

## 5    DISCUSSION & CONCLUSION

**On the use of automatic optimization for decision-making.**    We do not think that optimization algorithms should ever replace decision-makers, especially in contexts involving the life and death of individuals such as epidemics. However, we think optimization algorithms can provide decision-makers with useful insights by taking into account long-term effects of control policies. To this end, we believe it is important to consider a wide range of models and optimization algorithms to target a same control problem. This fosters a diversity of methods and reduces the negative impacts of model- and algorithm-induced biases. The EpidemiOptim toolbox is designed for this purpose: it provides a one-stop shop centralizing both the modeling and the optimization in one place.

**The future of EpidemiOptim.**    This toolbox is designed as a collaborative toolbox that will be extended in the future. To this day, it contains one population-based SEIR epidemiological models and three optimization algorithms. In the future, we plan on extending this library via collaborations with other research labs in epidemiology and machine learning. We plan on integrating agent-based epidemiological models that would allow a broader range of action modalities, and to add a variety of multi-objective algorithms from EAs and RL. We could finally imagine the organization of challenges, where optimization practitioners would compete on a set of epidemic control environments.

Although this toolbox focuses on the use of optimization algorithm to solve epidemic control policies, similar tools could be used in other domains. Indeed, just like epidemics, a large diversity of dynamical systems can be modeled models by ordinary differential equations (e.g. in chemistry, biology, economics, etc.). One could imagine spin-offs of EpidemiOptim for applications to other domains. Furthermore, Chen et al. (2018) recently designed a differentiable ODE solver. Such solvers could be integrated to EpidemiOptim to allow differential learning through the model itself.

**Conclusion.**    This paper introduces EpidemiOptim, a toolbox that facilitates collaboration between epidemiologists, optimization practitioners and others in the study of epidemic control policies to help decision-making. Of course, others have studied the use of optimization algorithms for the control of epidemics. We see this as a strength, as it shows the relevance of the approach and the importance of centralizing this research in a common framework that allows easy and reproducible comparisons and visualizations. Articulating our understanding of epidemics evolution with the societal impacts of potential control strategies on the population (death toll, physical and psychological traumas, etc) and on the economy will help improve the control of future epidemics.

ACKNOWLEDGMENTS

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

# A APPENDIX

## A.1 EPIDEMIOPTIM INTERFACE

Figure 4 presents the main interface of the EpidemiOptim library.

```python
from epidemioptim.environments.models import get_model
from epidemioptim.environments.cost_functions import get_cost_function
from epidemioptim.environments.gym_envs import get_env
from epidemioptim.optimization import get_algorithm
from epidemioptim.configs.get_params import get_params

config = 'dqn'

# Get the configuration
params = get_params(config_id=config)

# Get the epidemiological model
model = get_model(model_id=params['model_id'], params=params['model_params'])

# Get cost function
cost_function = get_cost_function(cost_function_id=params['cost_id'], params=params['cost_params'])

# Create the optimization problem as a Gym-like environment
env = get_env(env_id=params['env_id'], cost_function=cost_function, model=model, sim_horizon=params['sim_horizon'], seed=params['seed'])

# Get DQN algorithm parameterized by beta
algorithm = get_algorithm(algo_id=params['algo_id'], env=env, params=params)

# Run the training loop
algorithm.learn(num_train_steps=params['num_train_steps'])
```

Figure 4: Main running script of the EpidemiOptim library

## A.2 EXTENDED RELATED WORKS

Prior to the current COVID-19 pandemics, Yáñez et al. (2019) framed the problem of finding optimal intervention strategies for a disease spread as a reinforcement learning problem, focusing on how to design environments in terms of disease model, intervention strategy, reward function, and state representations. In Alamo et al. (2020), the CONCO-Team (CONtrol COvid-19 Team) provides a detailed SWOT analysis (Strengths, Weaknesses, Opportunities, Threats) and a roadmap that goes from the access to data sources to the final decision-making step, highlighting the interest of standard optimization methods such as Optimal Control Theory, Model Predictive Control, Multi-objective control and Reinforcement Learning. However, as argued in Shearer et al. (2020), a decision model for pandemic response cannot capture all of the social, political, and ethical considerations that impact decision-making. It is is therefore a central challenge to propose tools that can be easily used, configured and interpreted by decision-makers.

The aforementioned contributions provide valuable analyses on the range of methods and challenges involved in applying optimization methods to assist decision making during pandemics events. They do not, however, propose concrete implementations of such systems. Several recent contributions have proposed such implementations. Yaesoubi et al. (2020) applied an approximate policy iteration algorithm of their own (Yaesoubi & Cohen, 2016) on a SEIR model calibrated on the 1918 Influenza Pandemic in San Francisco. An interesting contribution of the paper was the development of a pragmatic decision tool to characterize adaptive policies that combined real-time surveillance data with clear decision rules to guide when to trigger, continue, or stop physical distancing interventions. Arango & Pelov (2020) applied a more standard RL algorithm (Double Deep Q-Network) on a SEIR model of the COVID-19 spread to optimize cyclic lock-down timings. The reward function was a fixed combination of a health objective (minimization of overshoots of ICU bed usage above an ICU bed threshold) and an economic objective (minimization the time spent under lock-downs). They compared the action policies optimized by the RL algorithm with a baseline on/off feedback control (fixed-rule), and with different parameters (reproductive number) of the SEIR model. Libin et al. (2020) used a more complex SEIR model with coupling between different districts and age groups. They applied the Proximal Policy Optimization algorithm to learn a joint policy that control the districts using a reward function quantifying the negative loss in susceptibles over one simulated week, with a constraint on the school closure budget. Probert et al. (2019) used Deep Q-Networks (DQN) and Monte-Carlo control on a stochastic, individual-based model of the 2001 foot-and-mouth

disease outbreak in the UK. An original aspect of their approach was to define the state at time $t$ using an image of the disease outbreak to capture the spatial relationships between farm locations, allowing the use of convolutional neural networks in DQN. Other contributions applied non-RL optimization methods such as deterministic rules (Tarrataca et al., 2020), stochastic approximation algorithms (Yaesoubi et al., 2020) or Bayesian optimization (Chandak et al., 2020). This latter paper also proposes a stochastic agent-based model called VIPER (Virus-Individual-Policy-EnviRonment) allowing to compare the optimization results on variations of the demographics and geographical distribution of population. Finally, Miikkulainen et al. (2020) proposed an original approach using Evolutionary Surrogate-assisted Prescription (ESP). In this approach, a recurrent neural network (the Predictor) was trained with publicly available data on infections and NPIs in a number of countries and applied to predicting how the pandemic will unfold in them in the future. Using the Predictor, an evolutionary algorithm generated a large number of candidate strategies (the Prescriptors) in a multi-objective setting to discover a Pareto front that represents different tradeoffs between minimizing the number of COVID-19 cases, as well as the number and stringency of NPIs (representing economic impact).

As we have seen, existing contributions widely differ in their definition of epidemiological models, optimization methods, cost functions, state and action spaces, as well as methods for representing the model decisions in a format suitable to decision-makers. Our approach aims at providing a standard toolbox facilitating the comparison of different configurations along these dimensions in order to assist decision-makers in the evaluation and the interpretation of the range of possible intervention strategies.

### A.3 ADDITIONAL METHODS

#### A.3.1 OPTIMIZATION ALGORITHMS

**The Deep Q-Network algorithm.** DQN was introduced in (Mnih et al., 2015) as an extension of the Q-learning algorithm (Watkins & Dayan, 1992) for policies implemented by deep neural networks. The objective is to train a Q-function to approximate the value of a given state-action pair $(s, a)$. This value can be understood as the cumulative measure of reward that the agent can expect to collect in the future by performing action $a_t$ now and following an optimal policy afterwards. In the Q-learning algorithm, the Q-function is trained by:

$$Q(s_t, a_t) \leftarrow Q(s_t, a_t) + \alpha[r_{t+1} + \gamma \max_a Q(s_{t+1}, a) - Q(s_t, a_t)].$$

Here, $\alpha$ is a learning rate, $[r_{t+1} + \gamma \max_a Q(s_{t+1}, a) - Q(s_t, a_t)]$ is called the temporal difference (TD) error with $r_{t+1} + \gamma \max_a Q(s_{t+1}, a)$ the target. Indeed, the value of the current state-action pair $Q(s_t, a_t)$ should be equal to the immediate reward $r_{t+1}$ plus the value of the next state $Q(s_{t+1}, a_{t+1})$ discounted by the discount factor $\gamma$. Because we assume we behave optimally after the first action, action $a_{t+1}$ should be the one that maximizes $Q(s_{t+1}, a)$. Once this function is trained, the agent simply needs to take the action that maximize the Q-value in its current state: $a^* = \max_a Q(s_t, a)$.

Deep Q-Network only brings a few mechanisms to enable the use of large policies based on deep neural network see Mnih et al. (2015) for details.

**Goal-parameterized DQNs.** Schaul et al. (2015) introduced Universal Value Function Approximators: the idea that Deep RL agents could be trained to optimize a whole space of cost functions instead of a unique one. In their work, cost functions are parameterized by latent codes called *goals*. An agent in a maze can decide to target any goal position within that maze. The cost function is then a function of that goal. To this end, the agent is provided with its current goal. In the traditional setting, the goal was concatenated to the state of the agent. In a recent work, Badia et al. (2020) used a cost function decomposed into two sub-costs. Instead of mixing the two costs at the cost/reward level and learning Q-functions from this signal, they proposed to learn separate Q-functions, one for each cost, and to merge them afterwards: $Q(s, a) = (1 - \beta) Q_1(s, a) + \beta Q_2(s, a)$. We use this same technique for our multi-objective problems.

**Using constraints.** In RL, constraints are usually implemented by adding a strong cost whenever the constraint is violated. However, it might be difficult for the Q-function to learn how to model these sudden jumps in the cost signal. Instead, we take inspiration from the work of Badia et al. (2020). We train a separate Q-function for each cost, where the associated reward is -1 whenever

the constraint is violated. This Q-function evaluates how many times the constraint is expected to be violated in the future when following an optimal policy. Once this is trained, we can use it to filtrate the set of admissible actions: if $Q_{\text{constraint}}(s, a) > 1$, then we expect action $a$ to lead to at least 1 constraint violation. Once these actions are filtered, the action is selected according to the usual maximization of the traditional Q-function.

**NSGA-II.** Multi-objective algorithms do not require the computation of an aggregated cost but use all costs. The objective of such algorithms is to obtain the best Pareto front. NSGA-II (Deb et al., 2002) is a state-of-the-art multi-objective algorithm based on EAs (here a genetic algorithm). Starting from a parent population of solutions, parents are selected by a tournament selection involving the rank of the Pareto front they belong to (fitness) and a crowding measure (novelty). Offspring are then obtained by cross-over of the parent solutions and mutations of the resulting parameters. The offspring population is then evaluated and sorted in Pareto fronts. A new parent population is finally obtained by selecting offspring from the higher-ranked Pareto fronts, prioritizing novel solutions as a second metric (low crowding measures).

### A.3.2 Epidemiological model

Figure 5 presents the schematic view of the mechanistic model behind the SEIRAH model from Prague et al. (2020) (left), and its underlying system of differential equations (right). See details in the original paper (Prague et al., 2020).

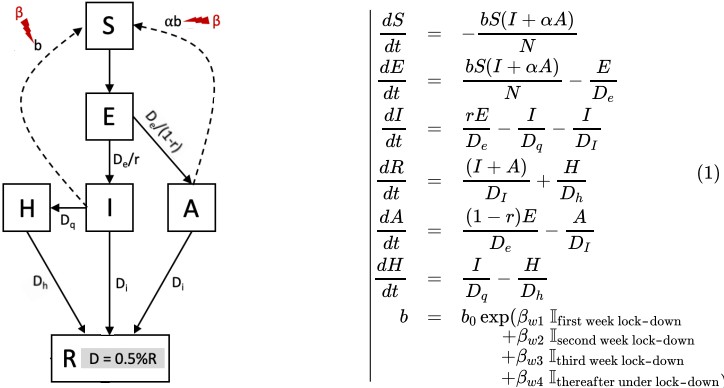

Figure 5: Description of the SEIRAH epidemiological model. Adapted from Prague et al. (2020)

To account for uncertainty in the model parameters, we use a distribution of models. At each episode, the epidemiological model is sampled from that distribution. The transition function is thus conditioned on a latent code (the model parameters) that is unknown to the agent. This effectively results in a stochastic transition from the viewpoint of the agent. To build the distribution of models, we assume a normal distributions for each of the model parameters, using either the standard deviation from the model inversion for parameters estimated in Prague et al. (2020) or $10\%$ of the mean value for other parameters. Those values are available in appendix Figure 6 features for reproducibility of the experiment. We further add a uniformly distributed delay between the epidemic onset and the start of the learning episode (uniform in [0, 21 days]). This delay models the reaction time of the decision authorities. This results in the distribution of models shown in Figure 7.

**Epidemiological model and economic cost function parameters.** Table 1 lists the parameters of the epidemiological model (from Prague et al. (2020)) and the economic cost function (from the National Institute of Statistics and Economical Analysis Insee (2020) and Havik et al. (2014)).

**Epidemiological model and cost function parameters.** Table 1 presents the list of parameters used in the epidemiological model (top) and the economic cost function (bottom).

| Param. | Interpretation | Value "Ile-de-France" |
|---|---|---|
| $b_0$ | Transmission rate of ascertained cases before lock-down | 2.23 |
| $r$ | Ascertainement rate | 0.043 |
| $\alpha$ | Ratio of transmission between $A$ and $I$ | 0.55 |
| $D_e$ | Latent (incubation) period (days) | 5.1 |
| $D_I$ | Infectious period (days) | 2.3 |
| $D_q$ | Duration from $I$ onset to $H$ (days) | 0.36 |
| $D_h$ | Hospitalization period (days) | 30 |
| $N$ | Population size | 12,278,210 |
| $\beta$ | Lock-down effects first ($\beta_{w1}$), second ($\beta_{w2}$), third ($\beta_{w3}$) and thereafter weeks ($\beta_{w'}$) | (-0.11, -0.50, -1.36, -1.46) |
| $S_0$ | Initial number of susceptible | $N - E_0 - I_0 - R_0 - A_0 - H_0$ |
| $E_0$ | Initial number of exposed | 5004 |
| $I_0$ | Initial number of ascertained | 16 |
| $R_0$ | Initial number of removed | 0 |
| $A_0$ | Initial number of non ascertained | 356 |
| $H_0$ | Initial number of hospitalized | 4 |
| $D_0$ | Initial number of death | 0 |
| $K_0$ | Initial capital stock in millions euros | 1,388,912 |
| $L_0$ | Number of employed individuals in millions | 4.58 |
| $\lambda$ | Activity/employment rate (%) | 37.4 |
| $Y_0$ | Initial GDP in million euros / year | 424,474 |
| $A$ | Exogenous technical progress | 867 |
| $u(t)$ | Level of partial unemployment during lock-down (%) | 50 |
| $\gamma_k$ | Capital elasticity | 0.37 |

Table 1: Parameter values for the *Île-de-France* region for the SEIRAH epidemiological model and the economy cost function.

|  | $E_0$ | $b$ | $\beta_{1w}$ | $\beta_{2w}$ | $\beta_{3w}$ | $\beta_{4w}$ | $D_q$ | $D_e$ | $D_h$ | $D_i$ | $\alpha$ | $r$ |
|---|---|---|---|---|---|---|---|---|---|---|---|---|
| $E_0$ | 52492 | -0,024 | -0,144 | 0,024 | 0,020 | 0,008 | 0,009 | 0 | 0 | 0 | 0 | 0 |
| $b$ | -0,024 | 0,00001 | 0 | 0 | 0 | 0 | 0 | 0 | 0 | 0 | 0 | 0 |
| $\beta_{1w}$ | -0,144 | 0 | 0,004 | 0 | 0 | 0 | 0 | 0 | 0 | 0 | 0 | 0 |
| $\beta_{2w}$ | 0,024 | 0 | 0 | 0,003 | 0 | 0 | 0 | 0 | 0 | 0 | 0 | 0 |
| $\beta_{3w}$ | 0,020 | 0 | 0 | 0 | 0,003 | 0 | 0 | 0 | 0 | 0 | 0 | 0 |
| $\beta_{4w}$ | 0,008 | 0 | 0 | 0 | 0 | 0,0002 | 0 | 0 | 0 | 0 | 0 | 0 |
| $D_q$ | 0,009 | 0 | 0 | 0 | 0 | 0 | 0,0001 | 0 | 0 | 0 | 0 | 0 |
| $D_e$ | 0 | 0 | 0 | 0 | 0 | 0 | 0 | 0,26 | 0 | 0 | 0 | 0 |
| $D_h$ | 0 | 0 | 0 | 0 | 0 | 0 | 0 | 0 | 9 | 0 | 0 | 0 |
| $D_i$ | 0 | 0 | 0 | 0 | 0 | 0 | 0 | 0 | 0 | 0,05 | 0 | 0 |
| $\alpha$ | 0 | 0 | 0 | 0 | 0 | 0 | 0 | 0 | 0 | 0 | 0,003 | 0 |
| $r$ | 0 | 0 | 0 | 0 | 0 | 0 | 0 | 0 | 0 | 0 | 0 | 0,00002 |

Figure 6: Variance-covariance matrix for parameters of the epidemiological model.

### A.4 Additional results

### A.4.1 A distribution of models

Figure 7 presents the evolution of the SEIRAH model states for a few models sampled from the distribution of models described in 4.1. Here, no lockdown is enforced.

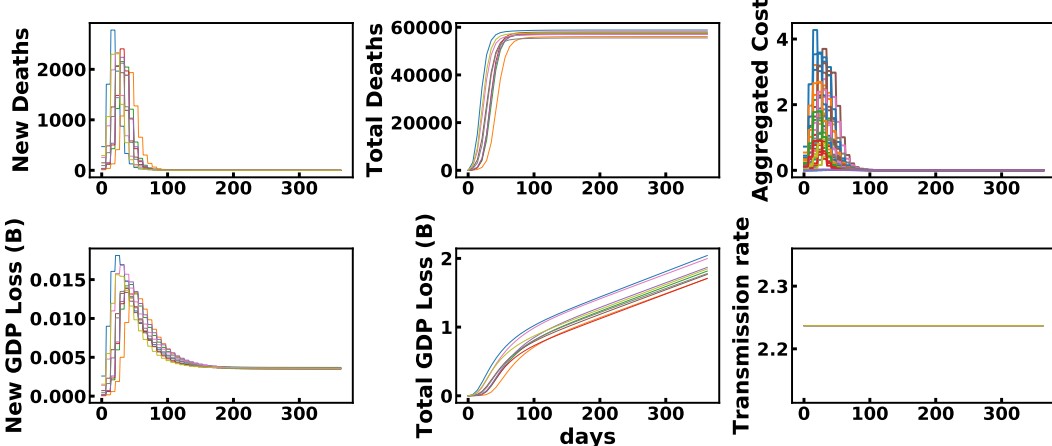

Figure 7: 10 models sampled from the distribution of epidemiological models used in the case-study.

Now, let us illustrate in details the results of the case study. To this end, we provide several runs of the different algorithms to illustrate their strategies.

### A.4.2 DQN

We first present strategies found by DQN.

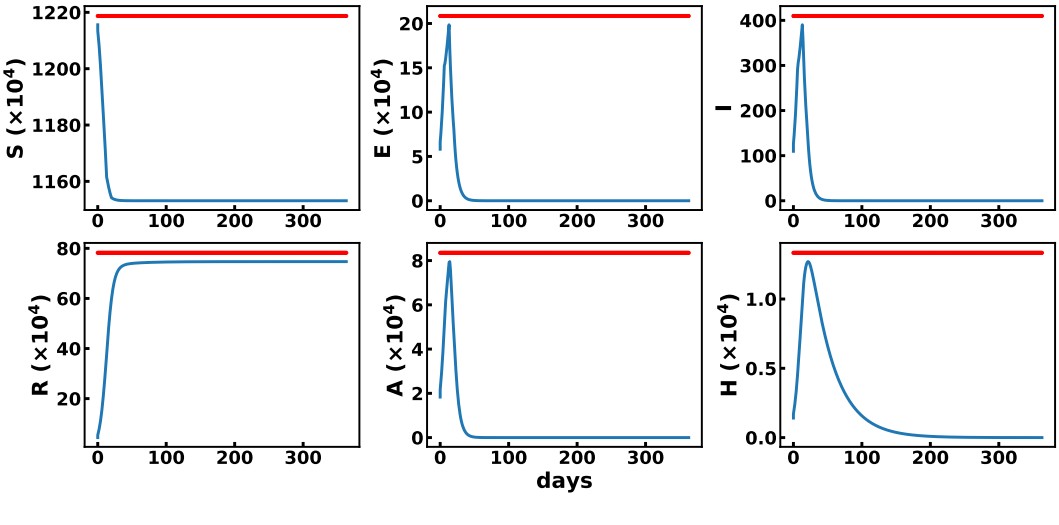

Eco cost: 155.16 B, Death Cost: 3513, Aggregated Cost: 5.40

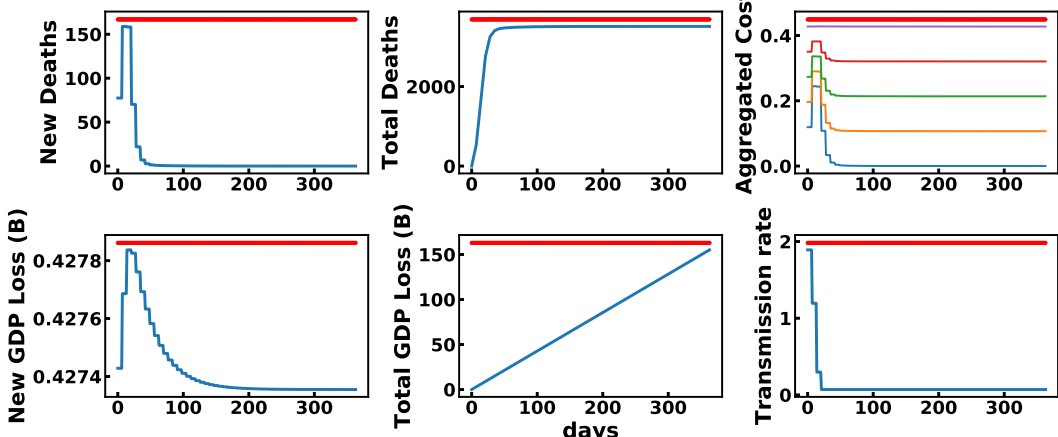

Figure 8: DQN trained with $\beta = 0$.. For one run, this figure shows the evolution of model states (above) and states relevant for optimization (below). The aggregated cost is shown for various values of $\beta$ in $[0, 0.25, 0.5, 0.75, 1]$. For $\beta = 0$, the agent only cares about the health cost, it always locks-down.

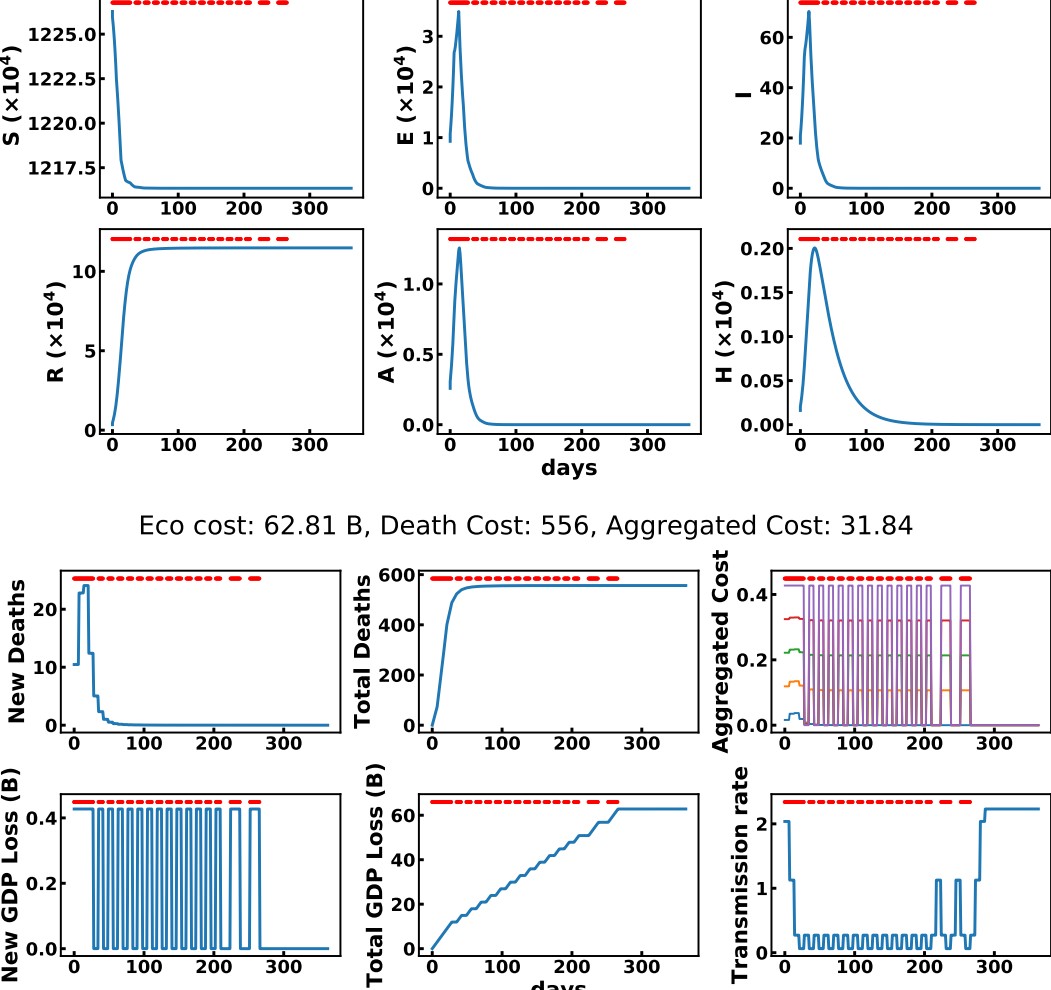

Figure 9: DQN trained with $\beta = 0.5$. Here the strategy is cyclical with a one or two weeks period. For one run, this figure shows the evolution of model states (above) and states relevant for optimization (below). The aggregated cost is shown for various values of $\beta$ in $[0, 0.25, 0.5, 0.75, 1]$.

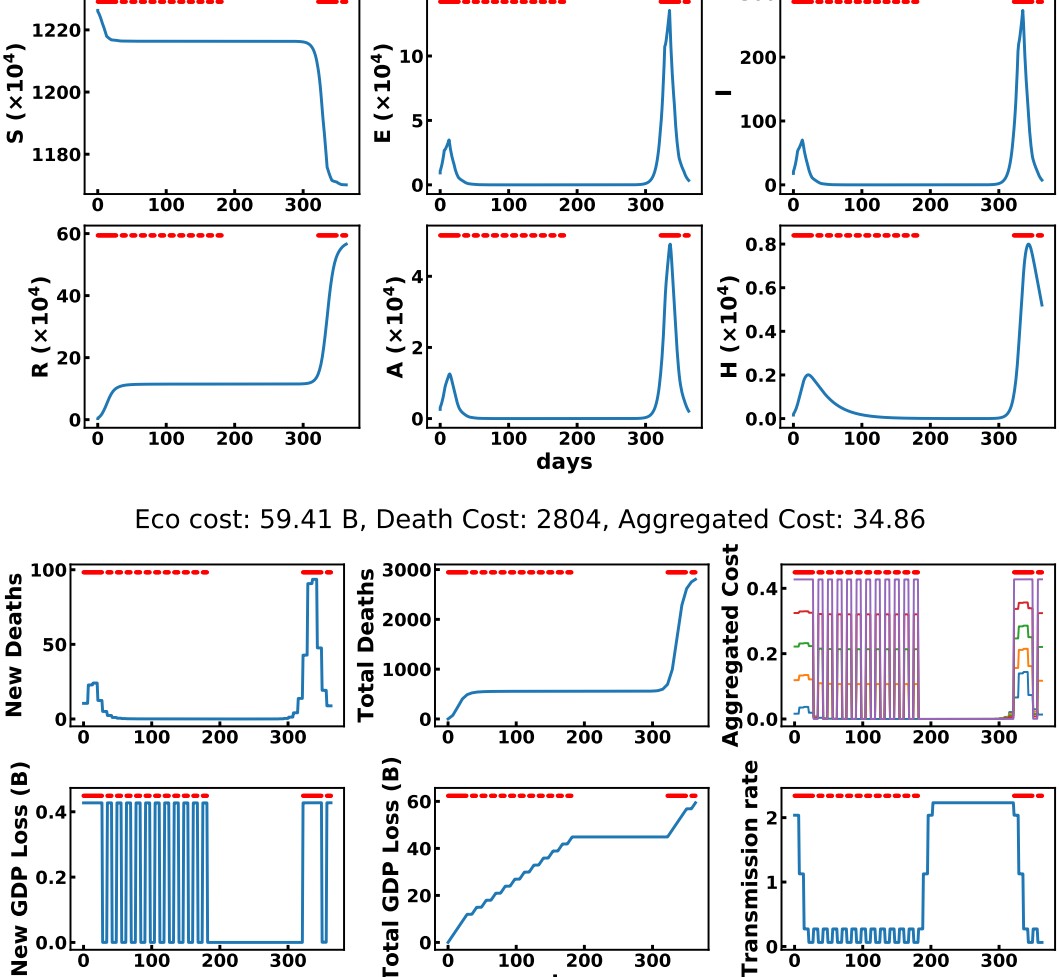

Figure 10: DQN trained with $\beta = 0.55$. Here the agent first lock-down to stop the first wave, but then stops the cyclical lockdown early which induces a second wave later, where the agent also reacts by a lock-down. For one run, this figure shows the evolution of model states (above) and states relevant for optimization (below). The aggregated cost is shown for various values of $\beta$ in $[0, 0.25, 0.5, 0.75, 1]$.

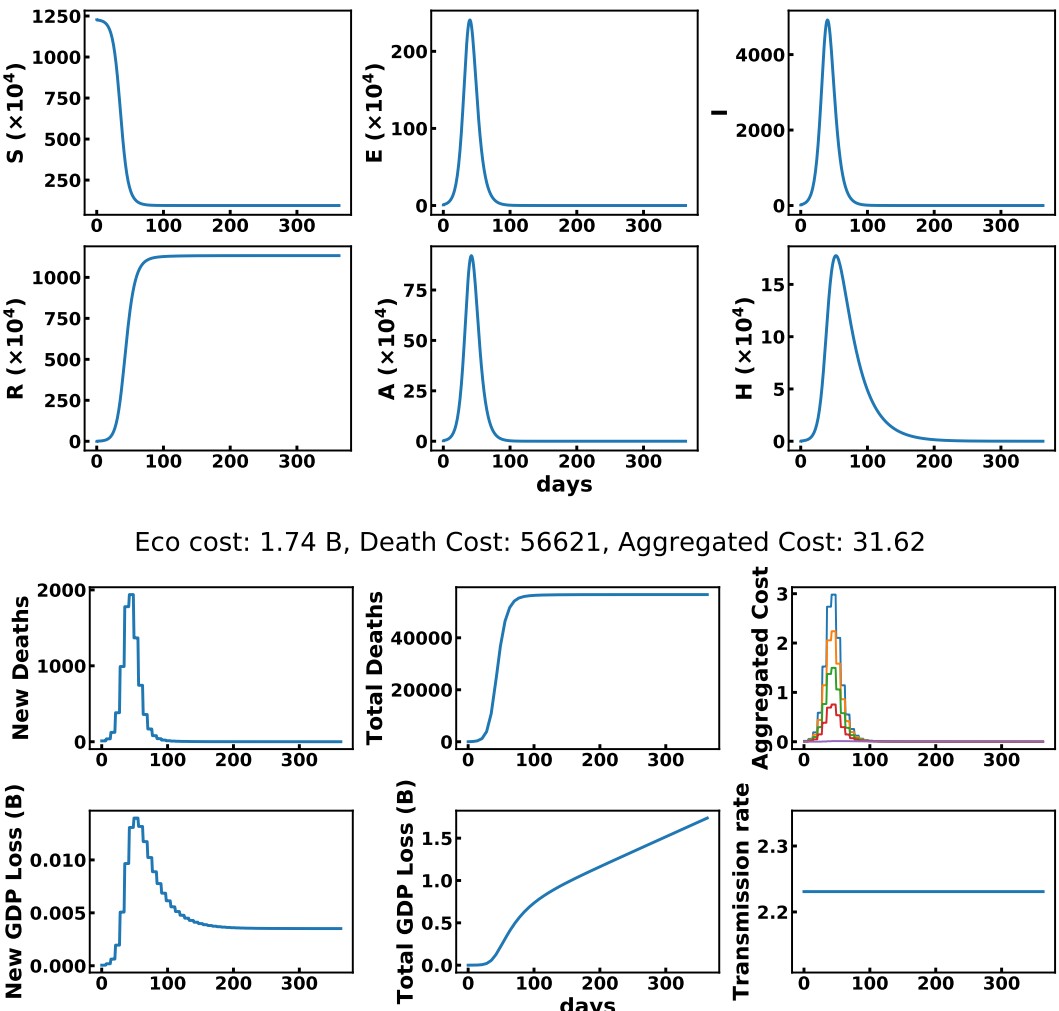

Figure 11: DQN trained with $\beta = 0.65$. Here the agent mostly cares about the economic cost, which results in a no-lockdown policy. For one run, this figure shows the evolution of model states (above) and states relevant for optimization (below). The aggregated cost is shown for various values of $\beta$ in $[0, 0.25, 0.5, 0.75, 1]$.

### A.4.3   GOAL-DQN WITHOUT CONSTRAINTS

Now we present a few strategies found by Goal-DQN without constraints.

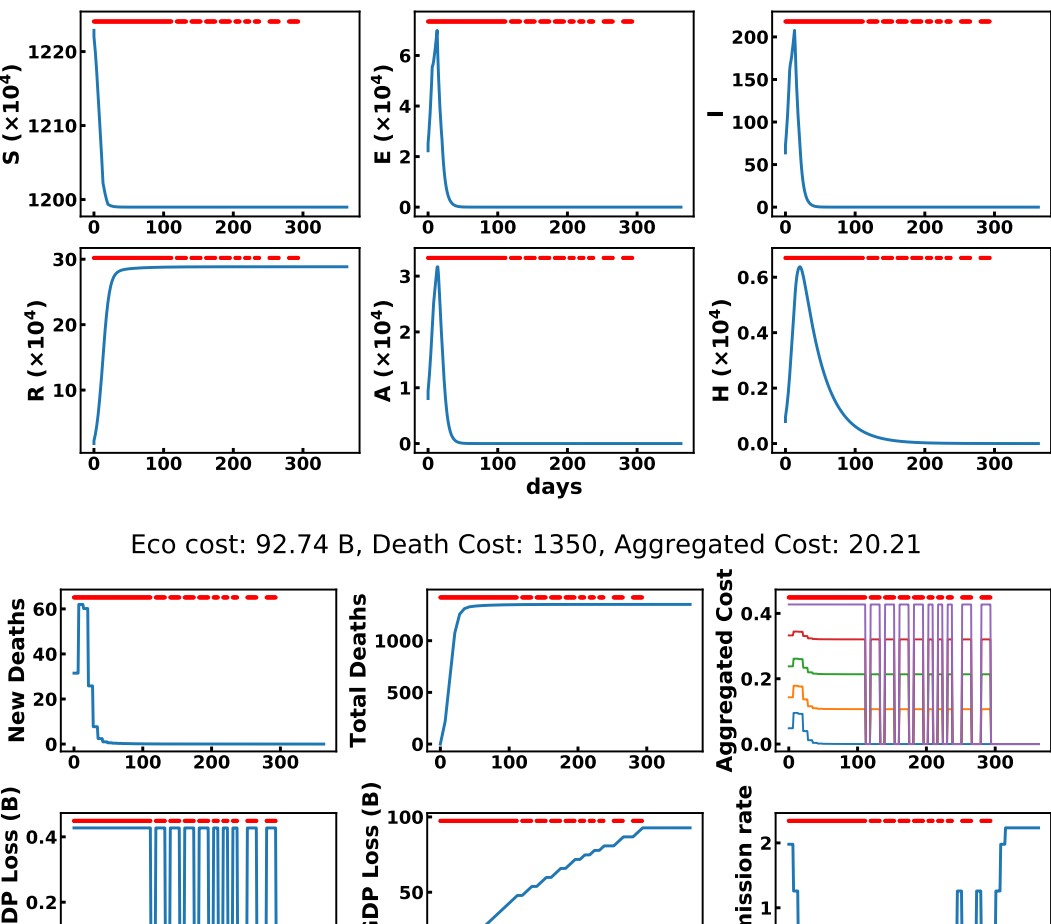

Figure 12: Goal-DQN without constraints evaluated in $\beta = 0.2$. Here the agent starts with a lasting lockdown, then pursues with cyclical lock-downs, which ensures the absence of second wave and, thus, results in low health cost but high economic cost. For one run, this figure shows the evolution of model states (above) and states relevant for optimization (below). The aggregated cost is shown for various values of $\beta$ in $[0, 0.25, 0.5, 0.75, 1]$.

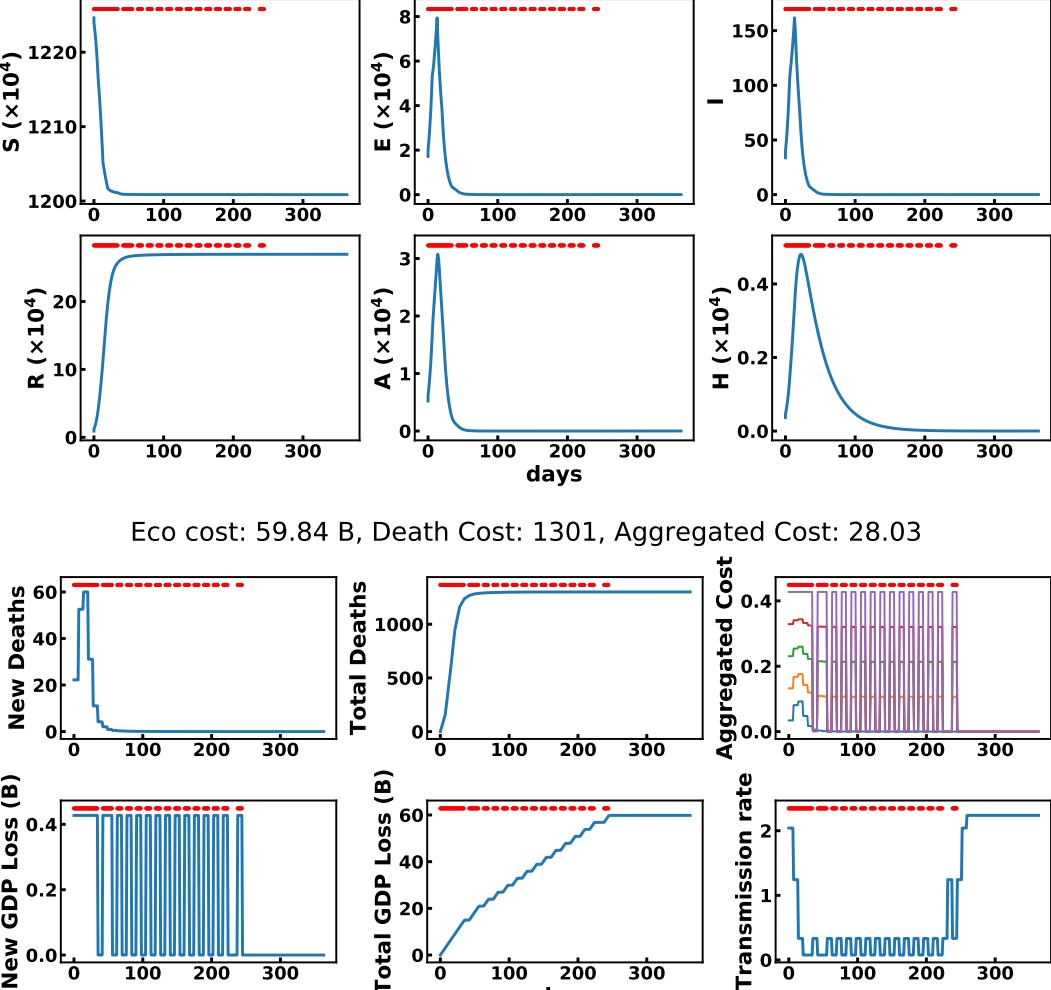

Figure 13: Goal-DQN without constraints evaluated in $\beta = 0.65$. Here we find a cyclical strategy equivalent to the one of shown with a DQN agent trained with $\beta = 0.5$. For one run, this figure shows the evolution of model states (above) and states relevant for optimization (below). The aggregated cost is shown for various values of $\beta$ in $[0, 0.25, 0.5, 0.75, 1]$.

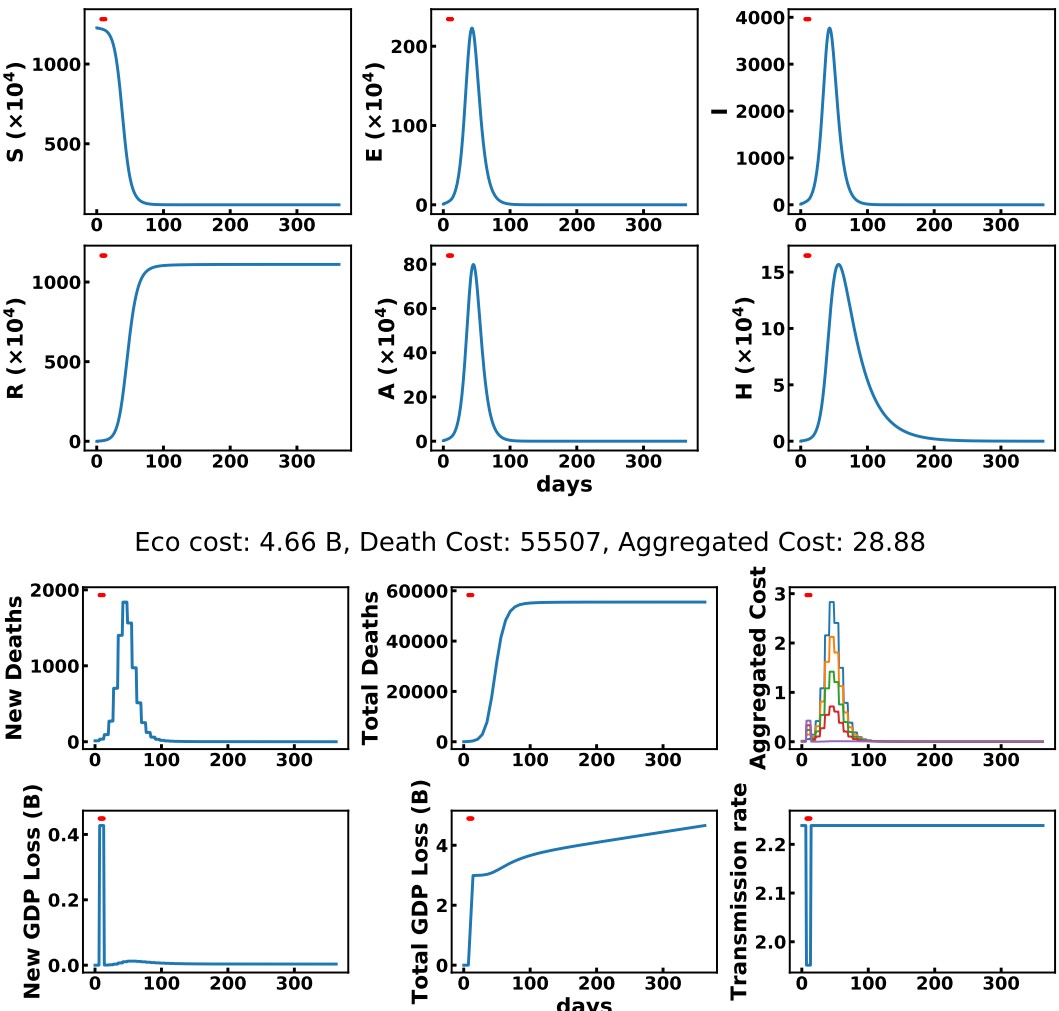

Figure 14: Goal-DQN without constraints evaluated in $\beta = 0.7$. Here the agent only use a lock-down at the very beginning of the epidemic. It is unclear whether this has an impact on the health cost at all. For one run, this figure shows the evolution of model states (above) and states relevant for optimization (below). The aggregated cost is shown for various values of $\beta$ in $[0, 0.25, 0.5, 0.75, 1]$.

### A.4.4 GOAL-DQN WITH CONSTRAINTS

Now we present a few strategies found by Goal-DQN with constraints, and study how it reacts to health and economic constraints.

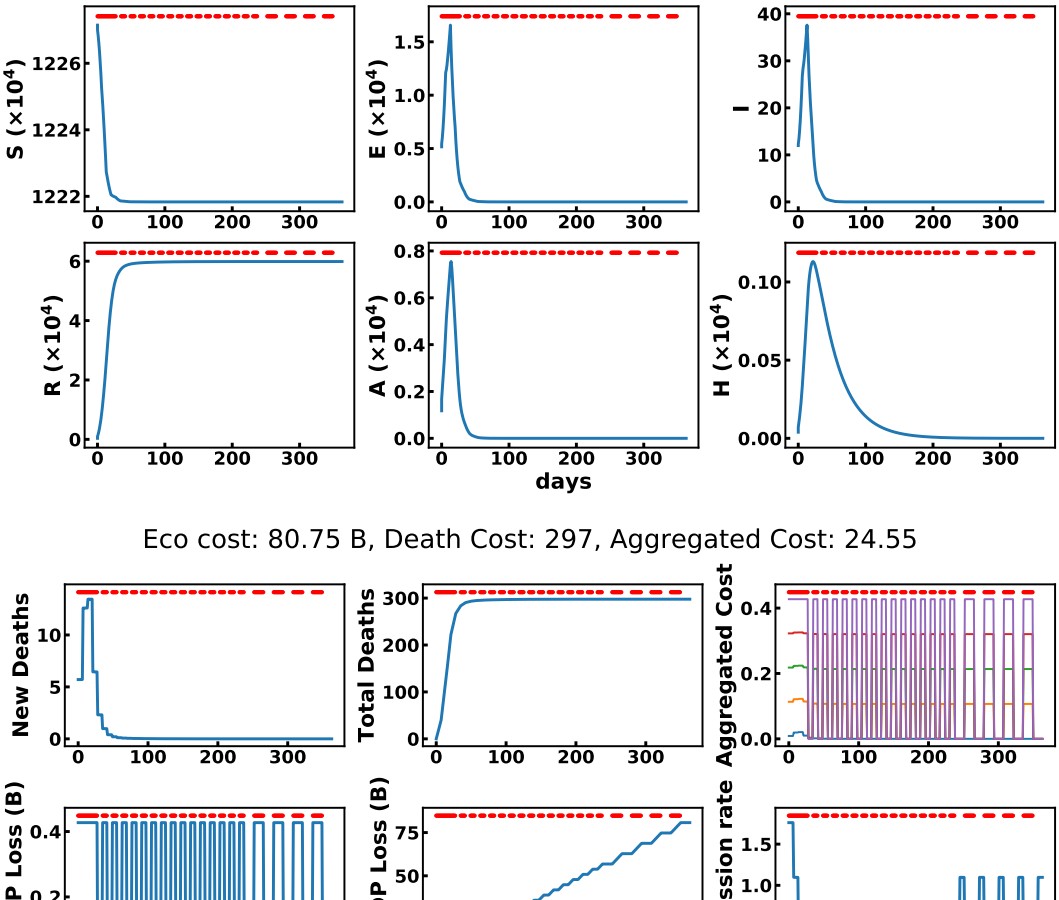

Figure 15: Goal-DQN with constraints evaluated in $\beta = 0.3, M_{\text{economic}} = 160B, M_{\text{health}} = 62000$ deaths. This boils down to no constraints as they are maximal values. This leads to a cyclical policy. For one run, this figure shows the evolution of model states (above) and states relevant for optimization (below). The aggregated cost is shown for various values of $\beta$ in $[0, 0.25, 0.5, 0.75, 1]$.

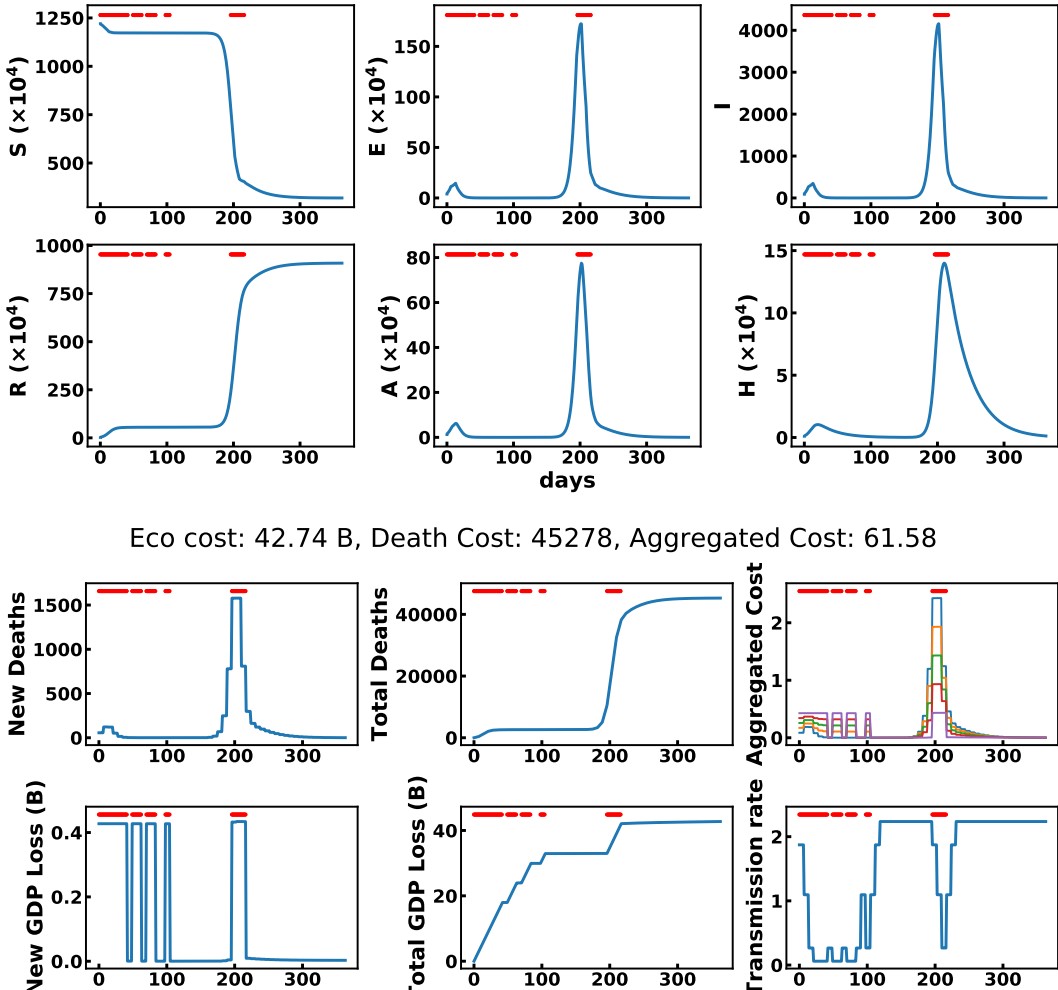

Figure 16: Goal-DQN with constraints evaluated in $\beta = 0.3, M_{\text{economic}} = 55B, M_{\text{health}} = 62000$ deaths. This is the same $\beta$ as Figure 15 (previous page). Now there is no constraint on the number of deaths but a strong constraint on the economic cost. The strategy is not cyclical anymore, as the resulting economic cost would be too high. This strategy stays below the economic constraint but still tries to minimize the health cost. For one run, this figure shows the evolution of model states (above) and states relevant for optimization (below). The aggregated cost is shown for various values of $\beta$ in $[0, 0.25, 0.5, 0.75, 1]$.

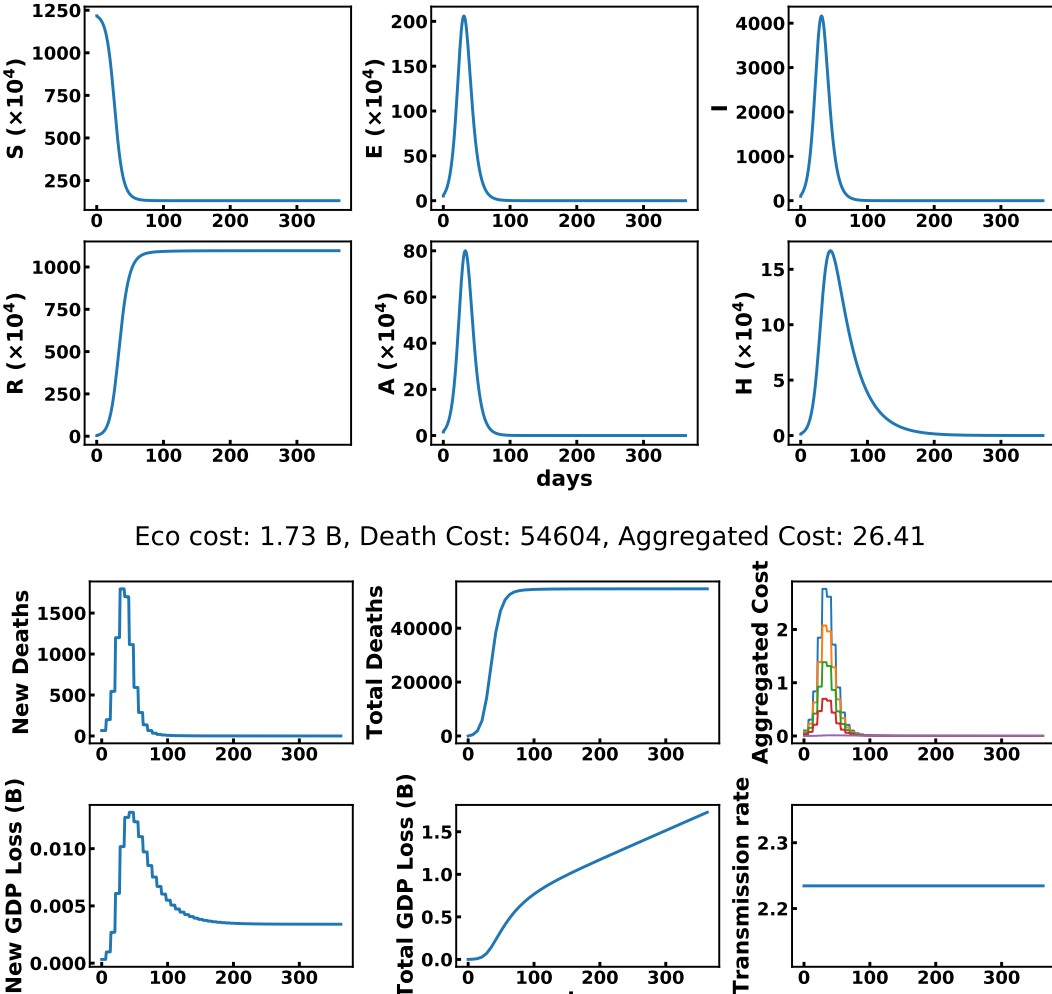

Figure 17: Goal-DQN with constraints evaluated in $\beta = 0.7, M_{\text{economic}} = 160B, M_{\text{health}} = 62000$ deaths. There is no constraint. As the balance favorizes the economic cost, the strategy does not implement any lock-down. For one run, this figure shows the evolution of model states (above) and states relevant for optimization (below). The aggregated cost is shown for various values of $\beta$ in $[0, 0.25, 0.5, 0.75, 1]$.

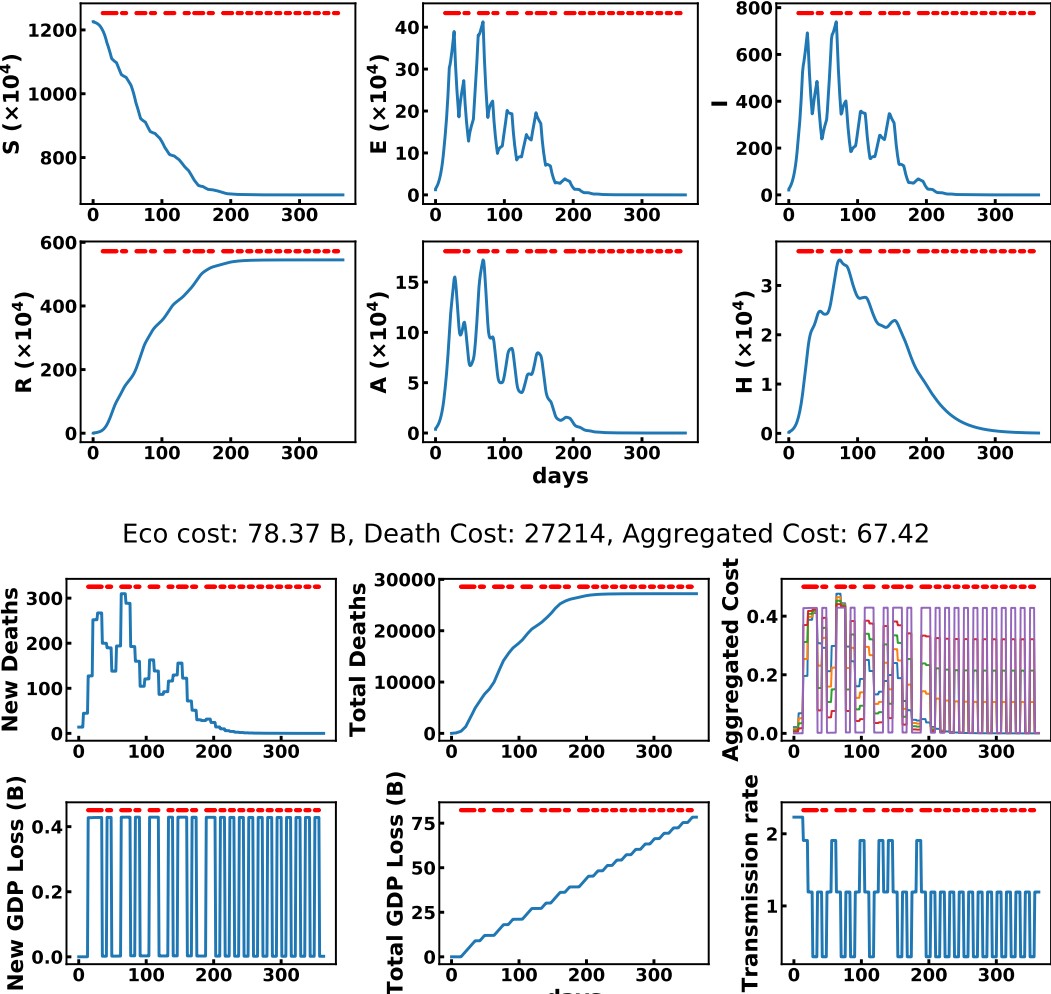

Figure 18: Goal-DQN with constraints evaluated in $\beta = 0.7, M_{\text{economic}} = 160B, M_{\text{health}} = 30500$ deaths. Now we have the same setup as the previous page (Figure 17), except that we have a strong constraint on the number of deaths. The resulting strategy respect the constraint while attempting to minimize the economic cost. For one run, this figure shows the evolution of model states (above) and states relevant for optimization (below). The aggregated cost is shown for various values of $\beta$ in $[0, 0.25, 0.5, 0.75, 1]$.

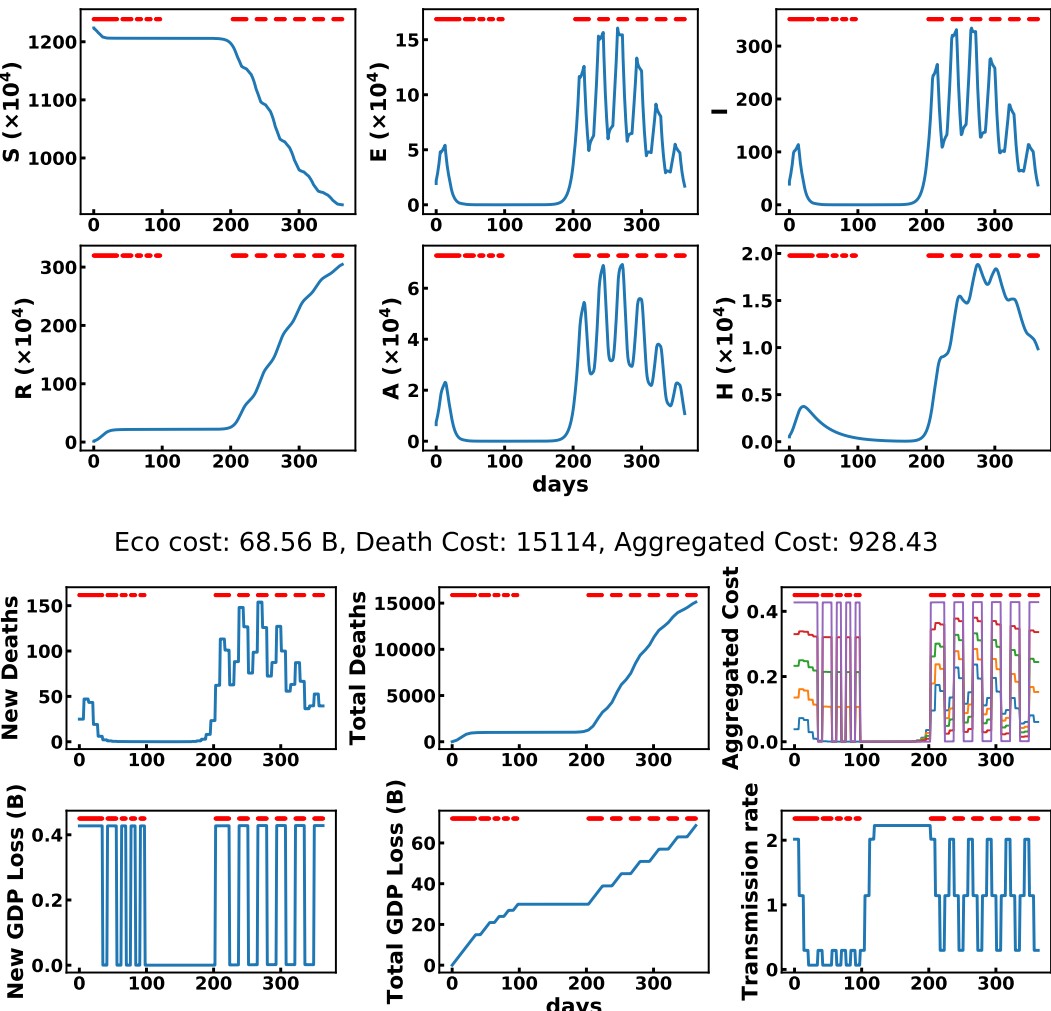

Figure 19: Goal-DQN with constraints evaluated in $\beta = 0.3, M_{\text{economic}} = 55B, M_{\text{health}} = 15000$ deaths. Here we have strong constraints on both economic costs. In that case, there is no good solution. This strategy respects the health constraint but violates the economic constraint. For one run, this figure shows the evolution of model states (above) and states relevant for optimization (below). The aggregated cost is shown for various values of $\beta$ in $[0, 0.25, 0.5, 0.75, 1]$.

### A.4.5 NSGA-II

Now we present a few strategies found by NSGA-II.

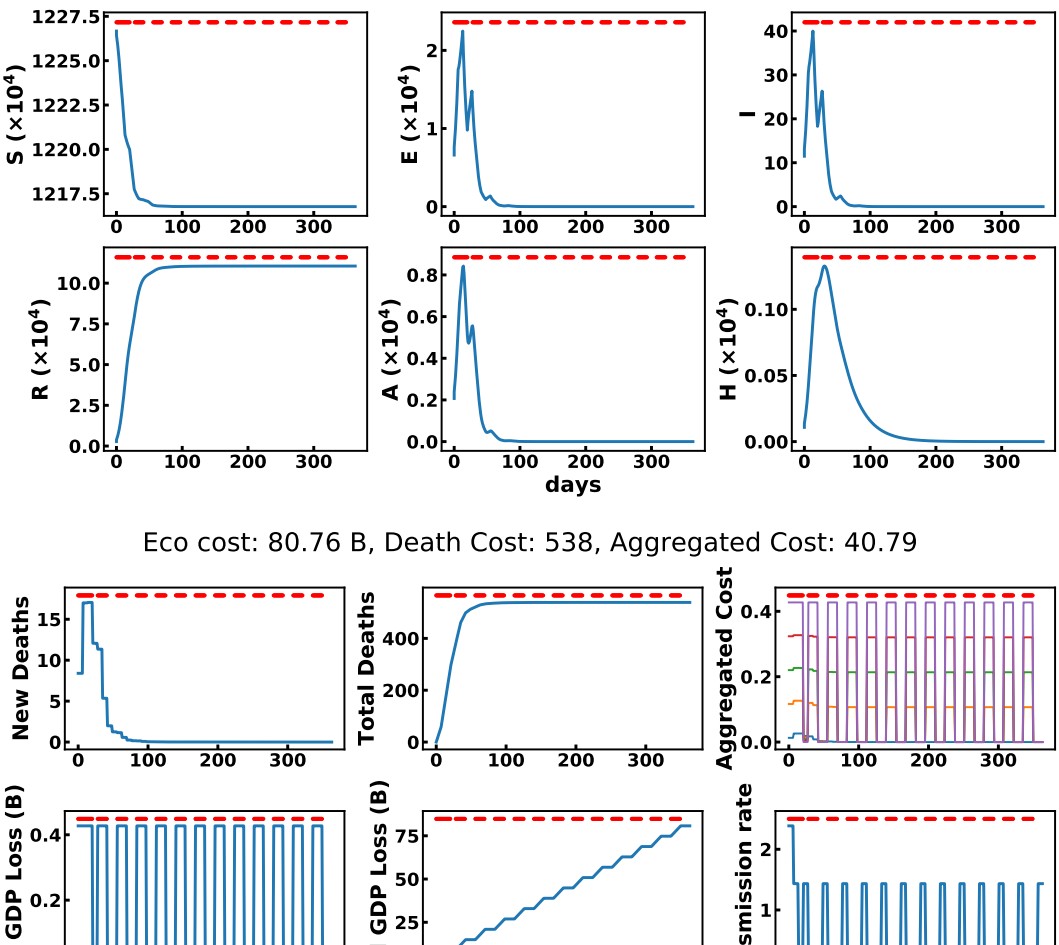

Figure 20: NSGA-II, the closest point to [8000 deaths, 60B] in the Pareto front. In this low health cost regime, NSGA-II finds a cyclical strategy with a period of 2 weeks. For one run, this figure shows the evolution of model states (above) and states relevant for optimization (below). The aggregated cost is shown for various values of $\beta$ in $[0, 0.25, 0.5, 0.75, 1]$.

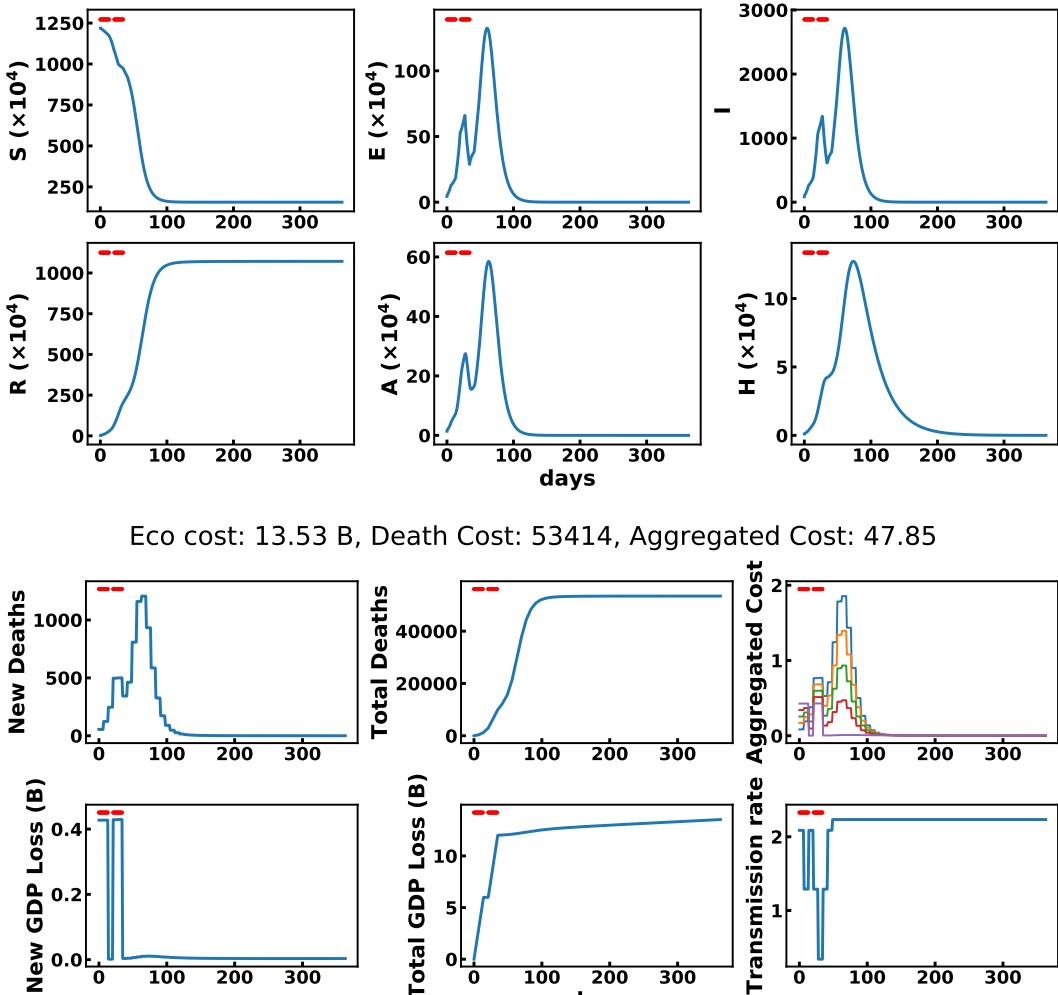

Eco cost: 13.53 B, Death Cost: 53414, Aggregated Cost: 47.85

Figure 21: NSGA-II, the closest point to [30000 deaths, 40B] in the Pareto front. Here we find that NSGA-II use alternative strategies depending on the epidemiological models it faces. The average of these strategies ends up close to 30000, although the two strategies either find high health costs (>50000) or low ones (<1000). These plots show the first alternative, where the strategy aims at breaking the first wave of the epidemic. See the other alternative in Figure 22. For one run, this figure shows the evolution of model states (above) and states relevant for optimization (below). The aggregated cost is shown for various values of $\beta$ in $[0, 0.25, 0.5, 0.75, 1]$.

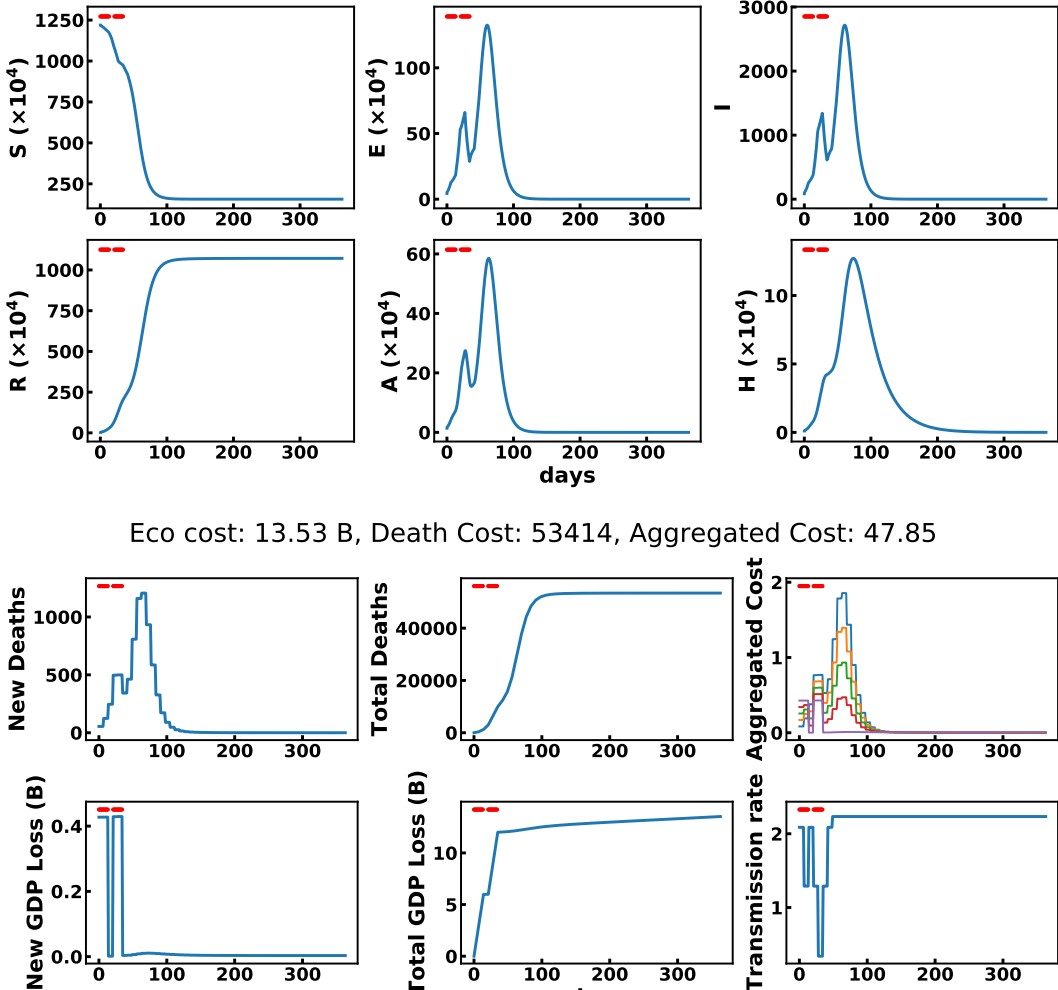

Figure 22: NSGA-II, the closest point to [30000 deaths, 40B] in the Pareto front. Here we find that NSGA-II use alternative strategies depending on the epidemiological models it faces. The average of these strategies ends up close to 30000, although the two strategies either find high health costs (>50000) or low ones (<1000). These plots show the second alternative, where the strategy is cyclical and achieved low health costs. See Figure 21 for the first alternative. For one run, this figure shows the evolution of model states (above) and states relevant for optimization (below). The aggregated cost is shown for various values of $\beta$ in $[0, 0.25, 0.5, 0.75, 1]$.

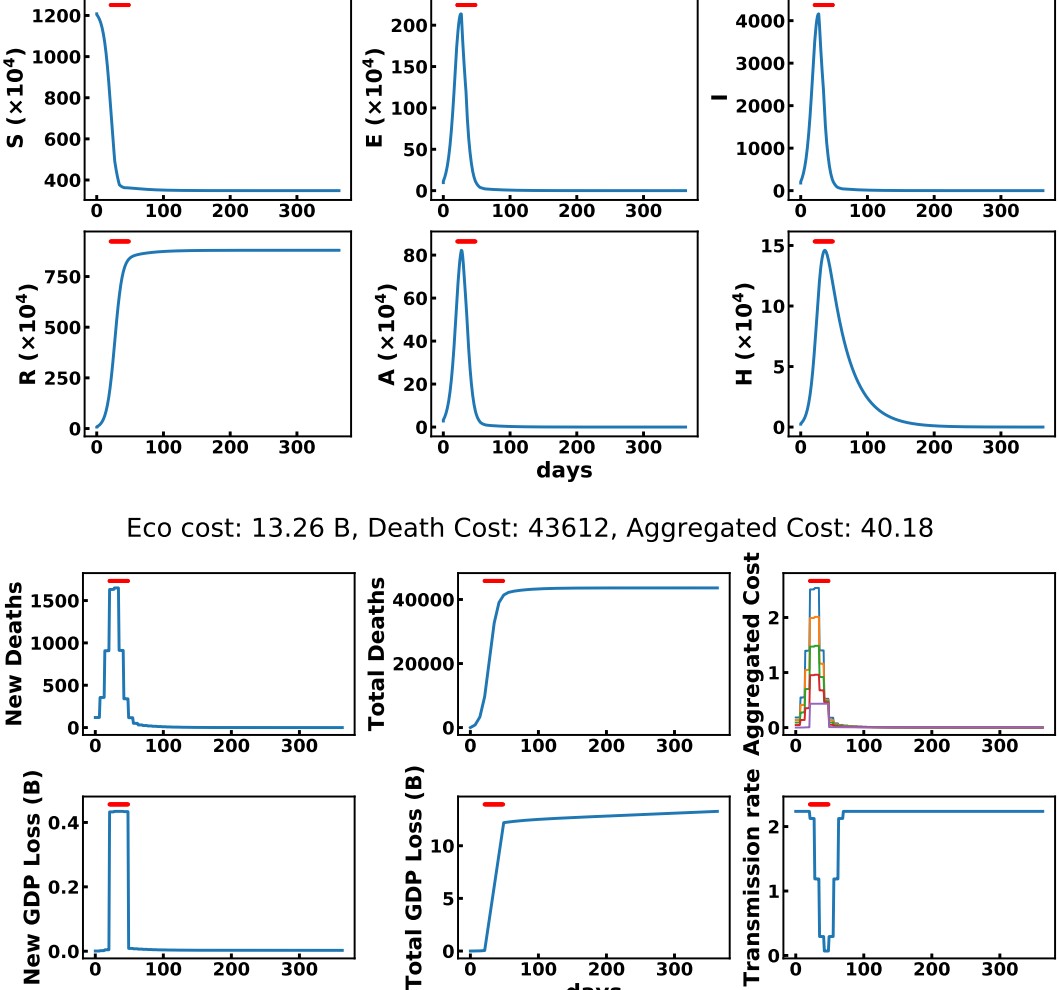

Figure 23: NSGA-II, the closest point to [45000 deaths, 20B] in the Pareto front. NSGA-II seem to find a robust strategy that consists in a single lock-down of a few weeks to break the first wave. For one run, this figure shows the evolution of model states (above) and states relevant for optimization (below). The aggregated cost is shown for various values of $\beta$ in $[0, 0.25, 0.5, 0.75, 1]$.

