# OpenReview forum: "EpidemiOptim: A Toolbox for the Optimization of Control Policies in Epidemiological Models"
_ICLR.cc/2021/Conference — Reject_

### Official Review · AnonReviewer3 · 2020-10-22
**Even if the work is interesting from a societal point of view, it is not enough to justify it as an ML study.**

**Rating:** 3
**Confidence:** 3

**Review:**

The authors provide a python tool able to model epidemics development as optimization problems. This allows easing the work of decision-makers when faced with the problem of deciding new lockdowns. The model has been applied to real-world data to evaluate the consequences, in terms of deaths and per-capita loss, of a new lockdown.

The paper presents an interesting study on the dynamics of the epidemic. In my opinion, the development of a tool is relevant for medical and decision-making studies but is not enough novel or significant for the ML field. I think that this kind of analysis is better suited for a more applicative venue, while the novelty provided in your work is not enough to justify a publication at ICLR.

The paper is clear and well written. I appreciate the use of a real case-study and the following analysis. I think that the only problem is that the venue chosen by the authors does not fit its purpose.


Questions:
- Did you also try your model on different datasets? I think that a more wide experimental campaing might improve the value of what have been proposed here
- What about other forms of prevention other than lockdowns? For instance, it is possible to model tracing or testing as prevention methods in your modeling? This would instead make the model more flexible and allow decisional organs to act in a more flexible way.
- Another interesting study might include the use of different methods for contagion prevention and their use in a joint or sequential manner in order to understand what are the policies over time which might be the most promising for a tradeoff healt/GDP.

----------------------------------------------------------------
After rebuttal: the paper seems interesting but, as I already mentioned and as other reviewers pointed out, the main concerns about this paper are novelty and relevance to the ML community.

---

> ### Author Response · Authors · 2020-11-18
> **Answer to R3**
>
> Here we answer specific comments and questions from R3.
>
> **About the study of other intervention modalities:**
> Studying other forms of intervention strategies is indeed a very interesting topic. However, to do this reliably, we need to be able to predict the impact of these intervention strategies on the epidemic. This means that we need data on the propagation of the epidemic in a region where this strategy was implemented. Some approaches use an LSTM-based model conditioned on intervention strategies to predict the dynamics of the epidemic. The prediction model is trained on various regions, where each region implemented specific intervention strategies. Based on this model, one might try to train an intervention policy. One potential pitfall in doing so is that the learning algorithm will try to implement intervention strategies in regions where it was never applied in real life. This leads to predictions that are out of the distribution of the training data. The “closing schools” intervention might have a different impact depending on the country and relying on the generalization of a recurrent network to predict its impact in a new country where it was never applied might lead to poor predictions of the epidemic dynamics, which in turn will lead to poor intervention strategies.  In our case, we only implemented a binary lockdown intervention modality, but we can be more confident about its impact on the epidemiological model because we already observed it in the past in the same regions (first wave of spring 2020). We just acquired recent data on the period of May 11 - October 30 that corresponds to a period of medium-level contact restrictions that was not as intensive as the lockdown periods (early spring and fall in France). This fresh data will help us to model the impact of such medium-level restrictions on the epidemic and will help us refine our model.

---

### Official Review · AnonReviewer2 · 2020-10-28
**Valuable starting point, but not yet sufficient contribution**

**Rating:** 4
**Confidence:** 4

**Review:**

This paper introduces OpenAI Gym environment for RL optimization of epidemic containment policies. The envirnoment currently contains an example SEIR model parameterized for COVID-19, along with a simple economic model to evaluate the lost productivity due to lockdowns. Some experimental results are shown where different deep learning algorithms are used to optimize intermittent lockdown policies.

On the positive side, connecting the epidemiological and ML communities is definitely an important goal. Developing open-source tools to make this interaction easier is valuable.

I'm not sure that this is an appropriate paper for ICLR though. It mostly takes a preexisting epidemiological model and exposes it in the OpenAI gym interface, without a strong research contribution. I believe that there are technical issues in the development of a platform for health policy optimization which are likely to result in research contributions, for example dealing with uncertainty in models/parameters, developing more efficient methods for multiobjective optimization, or providing explanations of policies (particularly since experts are unlikely to implement a RL policy verbatim, but rather try to synthesize its recommendations with other considerations/sources of information). I hope that the authors continue to refine this platform and tackle these or other issues in the future.

---

### Official Review · AnonReviewer1 · 2020-11-01
**Well written and easy to read paper. Unfortunately, work lacks novelty and real impact. Still seems premature and could benefit from more environments, agents, analysis and adoption.**

**Rating:** 3
**Confidence:** 4

**Review:**

**Summary**
This paper introduces a library that implements 1 infectious disease model and 3 agents in a gym environment. The authors provide analysis for a covid-19 lock down scenario using their library.

**Strengths**
The paper addresses an important and timely topic. The paper is easy to read and follow. The authors have open sourced their library and it seems to be well documented.

**Weaknesses**
The main weakness of the paper is the strength of the contributions. The authors’ main contributions are a gym environment for a specific epidemiological model.

Unfortunately, the novelty is somewhat lacking. As the authors mention in the conclusion section this line of work has been heavily researched. There are many infectious disease models and simulation environments already out there. E.g. a gym like interface has even been already open sourced earlier this year https://github.com/google/ml-fairness-gym/blob/master/environments/infectious_disease.py .

In terms of impact, the number of environments and agents is also quite limited at this stage. It’s also unclear if there is likely to be adoption by serious policy makers. Absent such impact statements it remains as one of many simulation frameworks out there.

**What could make the paper better**
a) Many more environments and agents need to be implemented such that this library has the potential to become the standard for infectious disease simulation.
b) A lot more analysis comparing agents and disease scenarios that truly unearth interesting scientific observations.
c) Real world impact statements of adoption by policy makers and governments.

---

### Author Response · Authors · 2020-11-18
**Main answer to all reviewers**

We would like to thank the reviewers for their feedback that will help refine this paper. Because the three reviews share most of their discussion points, we will write a common answer.

All reviewers noted the relevance and importance of the topic and two reviewers noted that the paper is clearly written and organized. However, all reviewers are concerned with the lack of novelty or technical contributions of this paper and two of them think that ICLR might not be the best venue for this type of work.

Regarding technical novelties and experiments, our paper introduces a novel variant of DQN that considers constraints via the training of additional Q-networks and samples its own constraints during training. It proposes a set of experiments investigating the optimization of lock-down policies with 4 different algorithms, where most past research only consider one or two. Besides, we show that they can be complementary (e.g. NSGA is efficient in low economic costs, DQN in low health costs). Although these two contributions may appear as limited with respect to the current ICLR standards (as noted by the reviewers), we argue below that the paper presents other forms of less usual but important contributions to the ML community.

The EpidemiOptim paper presents conceptual and organizational contributions. We formalize the problem of learning intervention strategies to mitigate epidemics propagations as a multi-objective optimization problem. Going further, we propose a modular view of the different components involved in such problematics:
* The learning environment should take the form of a Gym environment that contains three modules: an epidemiological model, a multi-objective cost function and action modalities.
* The optimization algorithm should be multi-objective.
In contrast to past approaches, this library does not simply present a novel Gym environment. It facilitates the addition of new modules by presenting a general interface that can be used to answer a vast list of research questions (e.g. impact of model uncertainty on optimal strategies, impact of the non-episodic nature of the underlying task, addition of new intervention modalities, etc.). In addition to the modules it contains, we provide relevant documentation as well as visualization and comparison tools to facilitate collaborative use. We agree that, in its current stage, the EpidemiOptim library contains a limited number of module implementations: 1 epidemiological models, 2 cost functions and 4 algorithms. However, we aim at presenting a collaborative library where anyone can easily contribute and add modules.

Because of the limited amount of novel technical contributions, R2 and R3 argue that this paper should not be published in an ML conference such as ICLR. We agree that this type of paper does not currently fit the current distribution of papers in ML venues. However, we believe that it should be the case because making progress in research is not only about the results but also about the tools we give ourselves to tackle interesting questions. This paper is about that: presenting the necessary tools to tackle the collaborative study of intervention strategies in the context of epidemics. The particular problem of the optimization of intervention strategies requires the attention and participation of optimization practitioners, most of whom attend ML conferences such as ICLR and are not familiar with the family of problems addressed by EpidOptim, which include novel challenging scientific dimensions and are societally important. We argue that accepting papers presenting toolboxes such as ours should be about judging their relevance in terms of the future organization of the research topic they tackle and judging their potential future impact in research and in the real world. In this scenario, accepting a toolbox paper would be a recommendation, stating that this toolbox presents a rational approach to organize research in this particular direction. Indeed, one cannot ask such a library to already be used by all the community and to already have impact in the real world (R1). If we submit this paper to one of the main ML venues, it is precisely because we think that, to lead to real world applications, we need the involvement of expert ML researchers focusing on this type of problems with the tools we propose to tackle them.


R1 and R3 suggest extending the set of analyses, to compare more agents and disease scenarios. R2 suggests investigating the handling of model uncertainty, new multi-objective algorithms or the explainability of intervention policies. R3 also suggests to study sequential combinations of policies or other intervention modalities. This demonstrates their interests in a diverse set of research questions that could be tackled via the EpidemiOptim framework. Of course, a single group cannot investigate them all, but a community of researchers working with a common and collaboratively-designed tools could.

---

### Decision · Program_Chairs · 2021-01-07
**Final Decision**

**Decision:**

Reject

**Comment:**

The reviewers agree that the contributions may not be relevant to the ML research community or perhaps are a poor fit for the venue, but otherwise find the work potentially useful and addressing a timely topic. Because the paper focuses on a simulation environment for existing epidemiological models, reviewers comment that the technical and methodological novelty is limited.